# Rapid protein assignments and structures from raw NMR spectra with the deep learning technique ARTINA

Piotr Klukowski [1] ✉, Roland Riek [1] ✉ & Peter Güntert [1,2,3] ✉

Nuclear Magnetic Resonance (NMR) spectroscopy is a major technique in structural biology with over 11,800 protein structures deposited in the Protein Data Bank. NMR can elucidate structures and dynamics of small and medium size proteins in solution, living cells, and solids, but has been limited by the tedious data analysis process. It typically requires weeks or months of manual work of a trained expert to turn NMR measurements into a protein structure. Automation of this process is an open problem, formulated in the field over 30 years ago. We present a solution to this challenge that enables the completely automated analysis of protein NMR data within hours after completing the measurements. Using only NMR spectra and the protein sequence as input, our machine learning-based method, ARTINA, delivers signal positions, resonance assignments, and structures strictly without human intervention. Tested on a 100-protein benchmark comprising 1329 multidimensional NMR spectra, ARTINA demonstrated its ability to solve structures with 1.44 Å median RMSD to the PDB reference and to identify 91.36% correct NMR resonance assignments. ARTINA can be used by non-experts, reducing the effort for a protein assignment or structure determination by NMR essentially to the preparation of the sample and the spectra measurements.

Studying structures of proteins and ligand-protein complexes is one of the most influential endeavors in molecular biology and rational drug design. All key structure determination techniques, X-ray crystallography, electron microscopy, and NMR spectroscopy, have led to remarkable discoveries, but suffer from their respective experimental limitations. NMR can elucidate structures and dynamics of small and medium size proteins in solution[1] and even in living cells[2]. However, the analysis of NMR spectra and the resonance assignment, which are indispensable for NMR studies, remain time-consuming even for a skilled and experienced spectroscopist. Attributed to this, the percentage of NMR protein structures in the Protein Data Bank (PDB) has decreased from a maximum of 14.6% in 2007 to 7.3% in 2021 (https://www.rcsb.org/stats). The problem has sparked research towards automating different tasks in NMR

structure determination[3,4], including peak picking[5-9], resonance assignment[10-12], and the identification of distance restraints[13,14]. Several of these methods are available as webservers[15,16]. This enabled semi-automatic[17,18] but not yet unsupervised automation of the entire NMR structure determination process, except for a very small number of favorable proteins[7,19].

The advance of machine learning techniques[20] now offers unprecedented possibilities for reliably replacing decisions of human experts by efficient computational tools. Here, we present a method that achieves this goal for NMR assignment and structure determination. We show for a diverse set of 100 proteins that NMR resonance assignments and protein structures can be determined within hours after completing the NMR measurements. Our method, *Arti*ficial *I*ntelligence for *N*MR *A*pplications, ARTINA (Fig. 1), combines machine

[1]Laboratory of Physical Chemistry, ETH Zurich, Vladimir-Prelog-Weg 2, 8093 Zurich, Switzerland. [2]Institute of Biophysical Chemistry, Goethe University Frankfurt, Max-von-Laue-Str. 9, 60438 Frankfurt am Main, Germany. [3]Department of Chemistry, Tokyo Metropolitan University, 1-1 Minami-Osawa, Hachioji 192-0397 Tokyo, Japan. ✉e-mail: piotr.klukowski@phys.chem.ethz.ch; roland.riek@phys.chem.ethz.ch; peter.guentert@phys.chem.ethz.ch

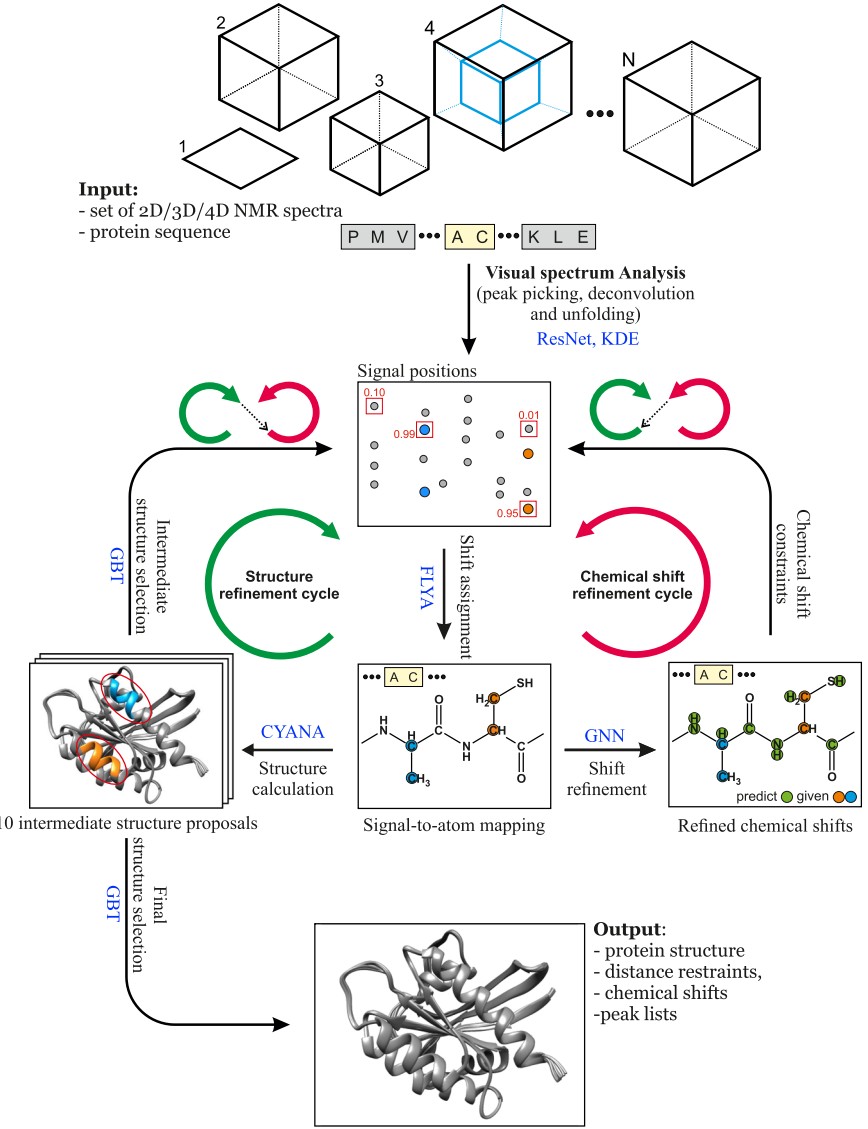

**Fig. 1 | The ARTINA workflow for automated NMR protein structure determination.** The flowchart presents the interplay between the main components of the automated protein structure determination workflow: Residual Neural Network (ResNet), FLYA automated chemical shift assignment, Graph Neural Network (GNN), Gradient Boosted Trees (GBT), and CYANA structure calculation.

learning for tasks that are difficult to model otherwise with existing algorithms−evolutionary optimization for resonance assignment with FLYA[12], chemical shift database searches for torsion angle restraint generation with TALOS-N[21], ambiguous distance restraints, network-anchoring and constraint combination for NOESY assignment[14,22] and simulated annealing by torsion angle dynamics for structure calculation with CYANA[23]. Machine learning is used in multiple flavors−deep residual neural networks[24] for visual spectrum analysis to identify peak positions (pp-ResNet) and to deconvolve overlapping signals (deconv-ResNet) in 25 different types of spectra (Supplementary Table 1), kernel density estimation (KDE) to reconstruct original peak positions in folded spectra, a deep graph neural network[25,26] (GNN) for chemical shift estimation within the refinement of chemical shift assignments, and a gradient boosted trees[27] (GBT) model for the selection of structure proposals.

A major challenge in developing ARTINA was the collection and preparation of a large training data set that is required for machine learning, because, in contrast to assignments and structures, NMR spectra are generally not archived in public data repositories. Instead, we were obliged to collect from different sources and standardize

complete sets of multidimensional NMR spectra for the assignment and structure determination of 100 proteins.

In the following work, we describe the algorithm, training and test data, and results of ARTINA automated structure determination, which are on par with those achieved in weeks or months of human experts' labor.

## Results

### Benchmark dataset

One of the major obstacles for developing deep learning solutions for protein NMR spectroscopy is the lack of a large-scale standardized benchmark dataset of protein NMR spectra. To date, published manuscripts presenting the most notable methods for computational NMR, typically refer to less than 50 2D/3D/4D NMR spectra in their experimental sections. Even the well-recognized CASD-NMR competition cannot serve as a major source of training data for deep learning, since only the NOESY spectra of 10 proteins were used in the last round of the event[28].

To make our study possible, we established a standardized benchmark of 1329 2D/3D/4D NMR spectra, which allows 100 proteins

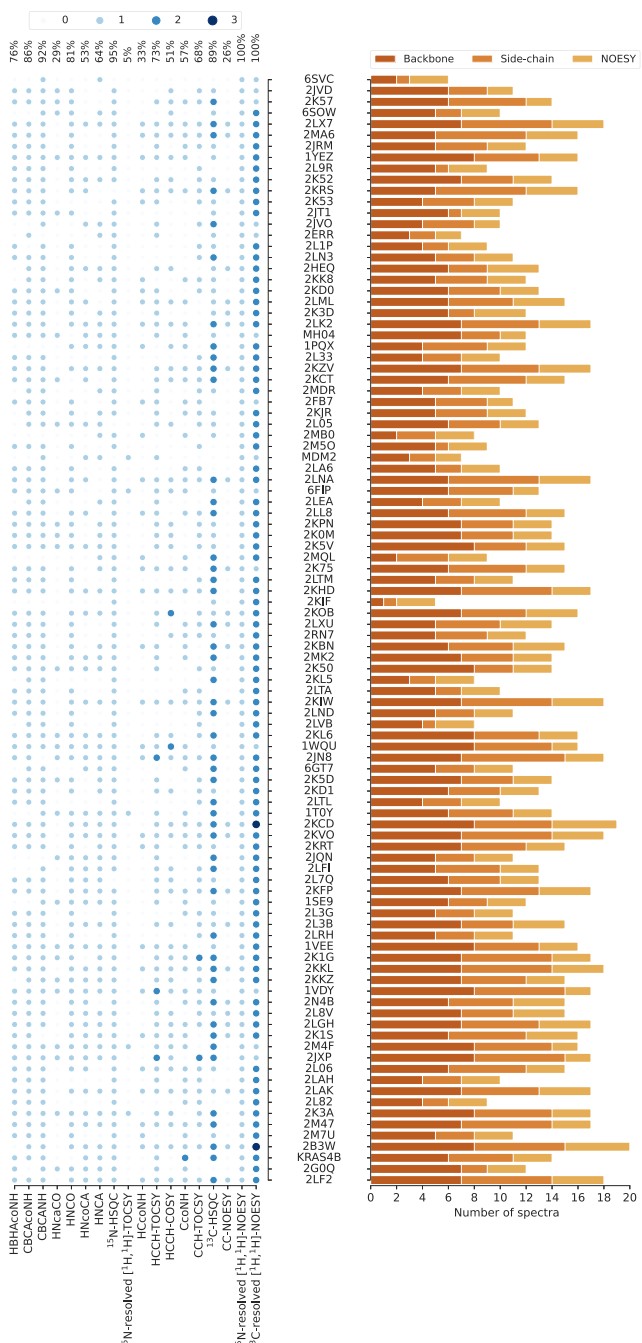

**Fig. 2 | NMR benchmark dataset content.** PDB codes (or names, MH04, MDM2, KRAS4B, if PDB code unavailable) of the 100 benchmark proteins are ordered by the number of residues. The histogram shows the number of spectra for backbone assignment, side-chain assignment, and NOE measurement. Spectrum types in each data set are shown by light to dark blue circles indicating the number of individual spectra of the given type. The percentages of benchmark records that contain a given spectrum type are given at the top. Spectrum types present in less than 5% of the data sets have been omitted.

## Automated protein structure determination

The accuracy of protein structure determination with ARTINA was evaluated in a 5-fold cross-validation experiment with the aforementioned benchmark dataset. Five instances of pp-ResNet and GBT were trained, each one using data from about 80% of the proteins for training and the remaining ones for testing. Since each protein was present exactly once in the test set, reported quality metrics were obtained directly in the cross-validation experiment, and no averaging between data splits was required. To deploy pp-ResNet and GBT models in our online system, we constructed an ensemble by averaging predictions of all 5 cross-validation models. The other models were trained only once using either generated data (deconv-ResNet, Supplementary Fig. 1) or BMRB depositions excluding all benchmark proteins (GNN, KDE).

In this experiment, we reproduced 100 structures in fully automated manner using only NMR spectra and the protein sequences as input. Since ARTINA has no tunable parameters and does not require any manual curation of data, each structure was calculated by a single execution of the ARTINA workflow. All benchmark datasets were analyzed by ARTINA in parallel with execution times of 4–20 h per protein.

All automatically determined structures, overlaid with the corresponding reference structures from the PDB, are visualized in Fig. 3, Supplementary Fig. 2, and Supplementary Movie 1. ARTINA was able to reproduce the reference structures with a median backbone root-mean-square deviation (RMSD) of 1.44 Å between the mean coordinates of the ARTINA structure bundle and the mean coordinates of the corresponding reference PDB structure bundle for the backbone atoms N, C$^\alpha$, C' in the residue ranges determined by CYRANGE[29] (Fig. 4a and Supplementary Table 4). ARTINA automatically identified between 459 and 4678 distance restraints (2198 on average over 100 proteins), which corresponds to 4.25–33.20 restraints per residue (Fig. 4b). This number is mainly influenced by the extent of unstructured regions and the quality of the NOESY spectra. In agreement with earlier findings[30], it correlates only weakly with the backbone RMSD to reference (linear correlation coefficient −0.38). As a more expressive validation measure for the structures from ARTINA, we computed a predicted RMSD to the PDB reference structure on the basis of the RMSDs between the 10 candidate structure bundles calculated in ARTINA (see "Methods", Fig. 5, and Supplementary Table 5). The average deviation between actual and predicted RMSDs for the 100 proteins in this study is 0.35 Å, and their linear correlation coefficient is 0.77 (Fig. 5). In no case, the true RMSD exceeds the predicted one by more than 1 Å.

Additional structure validation scores obtained from ANSSUR[31] (Supplementary Table 6), RPF[32] (Supplementary Table 7), and consensus structure bundles[33] (Supplementary Table 8) confirm that overall the ARTINA structures and the corresponding reference PDB structures are of equivalent quality. Energy refinement of the ARTINA structures in explicit water using OPALp[34] (not part of the standard ARTINA workflow) does not significantly alter the agreement with the PDB reference structures (Supplementary Table 9). The benchmark data set comprises 78 protein structures determined by the Northeast Structural Genomics Consortium (NESG). On average, ARTINA yielded structures of the same accuracy for NESG targets (median RMSD to reference 1.44 Å) as for proteins from other sources (1.42 Å).

On average, ARTINA correctly assigned 90.39% of the chemical shifts (Fig. 4c), as compared to the manually prepared assignments, including both "strong" (high-reliability) and "weak" (tentative) FLYA assignments[12]. Backbone chemical shifts were assigned more accurately (96.03%) than side-chain ones (86.50%), which is mainly due to difficulties in assigning lysine/arginine (79.97%) and aromatic (76.87%) side-chains. Further details on the assignment accuracy for individual amino acid types in the protein cores (residues with less than 20% solvent accessibility) are given in Supplementary Table 10. Assignments for core residues, which are important for the protein structure, are generally more accurate than for the entire protein, in particular

to be recalculated using their original spectral data (Fig. 2 and Supplementary Table 2). Each protein record in our dataset contains 5–20 spectra together with manually identified chemical shifts (usually depositions at the Biological Magnetic Resonance Data Bank, BMRB) and the previously determined ("ground truth") protein structure (PDB record; Supplementary Table 3). The benchmark covers protein sizes typically studied by NMR spectroscopy with sequence lengths between 35 and 175 residues (molecular mass 4–20 kDa).

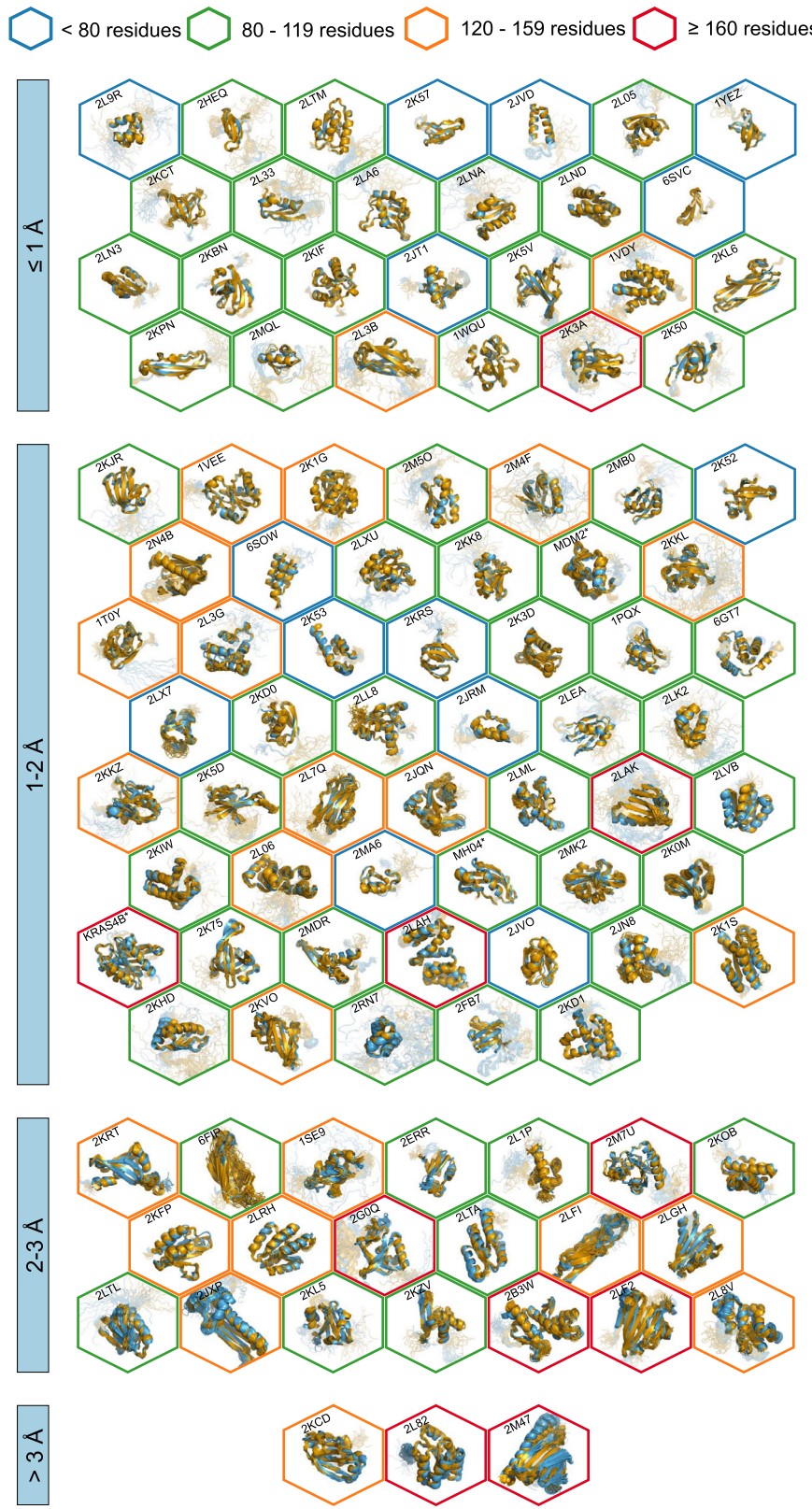

**Fig. 3 | 100 protein structures determined automatically by ARTINA (blue) overlaid with corresponding PDB depositions (orange).** The structures are aligned with the RMSD to reference range as indicated on the left and hexagonal frames color-coded by their size as indicated above. Structures with no corresponding PDB depositions are marked by an asterisk.

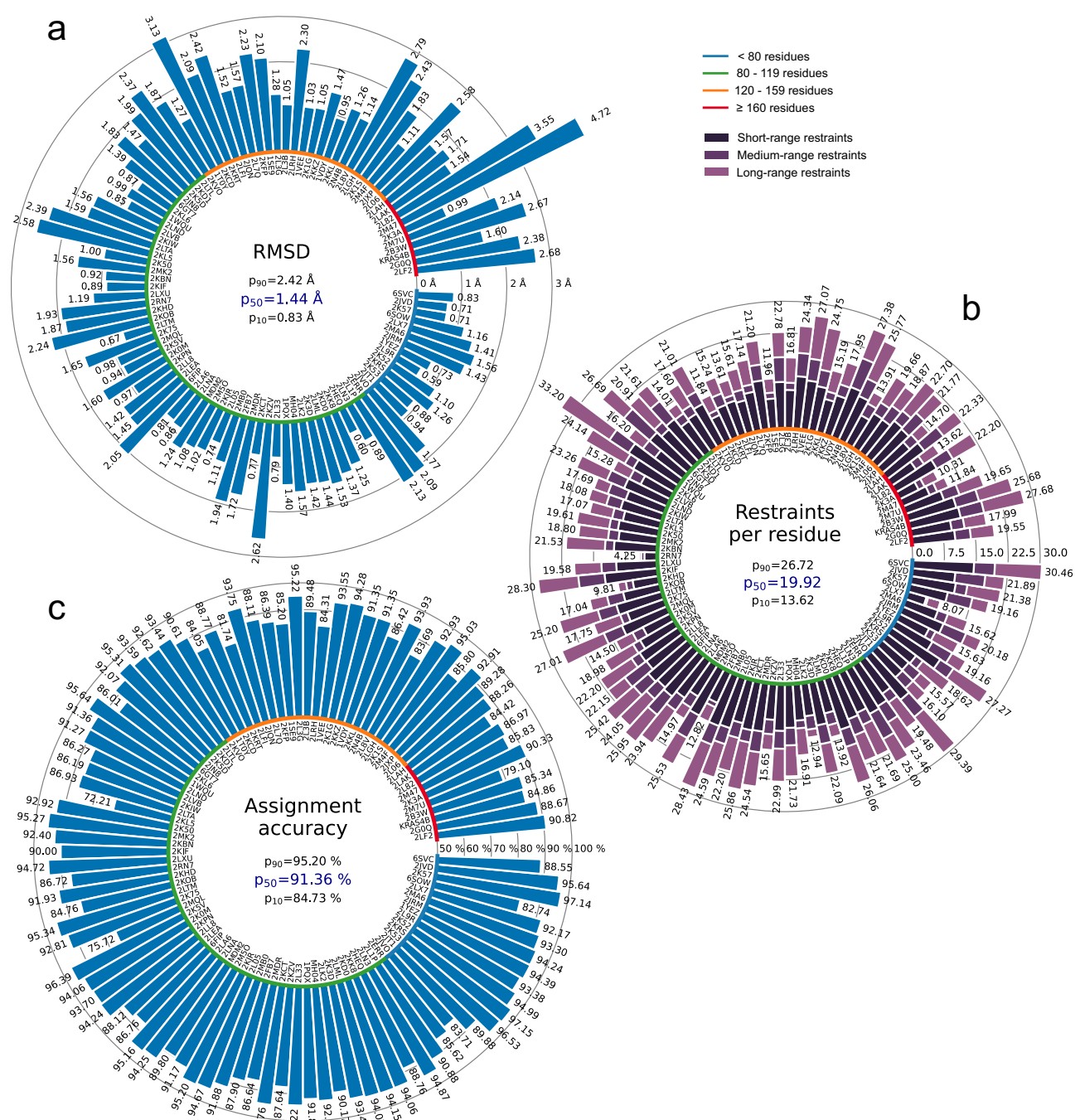

**Fig. 4 | Results of the automated structure determination of 100 proteins.**
**a** Backbone RMSD to reference. **b** Number of distance restraints per residue.
**c** Chemical shift assignment accuracy. Bars represent quantity values for benchmark proteins, identified by PDB codes (or protein names). Proteins are ordered by size, which is indicated by a color-coded circle. Values in the center of each panel are 10th, 50th, and 90th percentiles of values presented in the bar plot. Short/medium/long-range restraints are between residues $i$ and $j$ with $|i - j| \leq 1$, $2 \leq |i - j| \leq 4$, and $|i - j| \geq 5$, respectively.

for core Ala, Cys, and Asp residues, which show a median assignment accuracy of 100% over the 100 proteins. The lowest accuracies are observed for core His (83.3%), Phe (83.3%), and Arg (87.5%) residues. The three proteins with highest RMSD to reference, 2KCD, 2L82, and 2M47 (see below), show 68.2, 83.8, and 75.7% correct aromatic assignments, respectively, well below the corresponding median of 85.5%. On the other hand, the assignment accuracies for the methyl-containing residues Ala, Ile, Val are above average and reach a median of 100, 97.6, and 98.6%, respectively.

The quality of automated structure determination and chemical shift assignment reflects the performance of deep learning-based visual spectrum analysis, presented qualitatively in Figs. 6–7, Supplementary Fig. 3, and Supplementary Movies 2–4. In this experiment, our models (pp-ResNet, deconv-ResNet) automatically identified 1,168,739 cross-peaks with high confidence (≥0.50) in the benchmark spectra. All 1329 peak lists, together with automatically determined protein structures and chemical shift lists, are available for download.

## Error analysis

The largest deviations from the PDB reference structure were observed for the proteins 2KCD, 2L82, and 2M47, for which the pRMSD consistently indicated low accuracy (Fig. 5). Significant deviations are

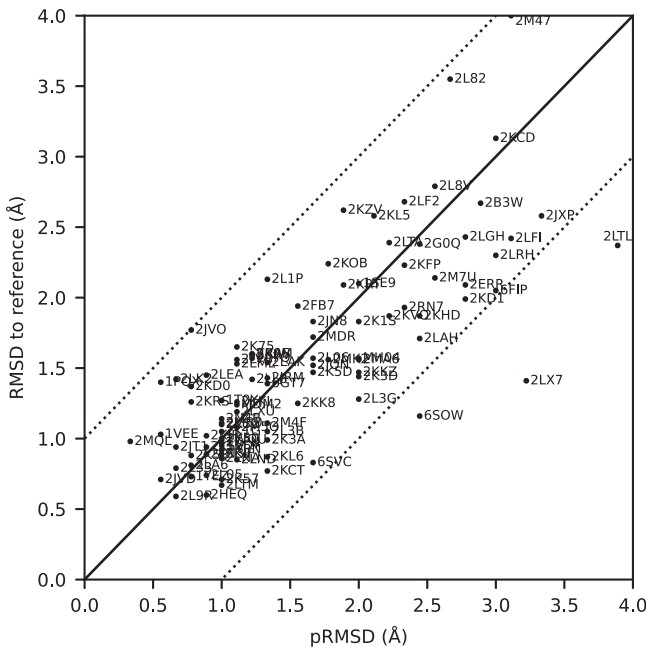

**Fig. 5 | Actual and predicted RMSD between ARTINA and reference PDB structures.** The predicted RMSD to reference (pRMSD) is calculated from the ARTINA results without knowledge of the reference PDB structure (see "Methods") and, by definition, always in the range of 0–4 Å. For comparability, actual RMSD values to reference are also truncated at 4 Å (protein 2M47 with RMSD 4.47 Å). The dotted lines represent deviations of ±1 Å between the two RMSD quantities.

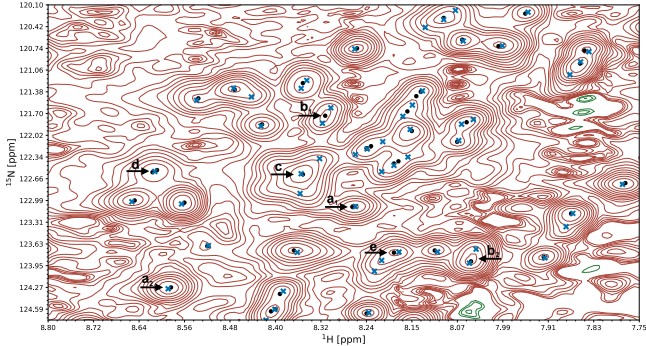

**Fig. 6 | Commonly occurring challenges in visual spectrum analysis.** A fragment of a $^{15}$N-HSQC spectrum of the protein 1T0Y is shown. Initial signal positions identified by the peak picking model pp-ResNet (black dots) are deconvolved by deconv-ResNet, yielding the final coordinates used for automated assignment and structure determination (blue crosses). **a₁, a₂** Initial peak picking marker position is refined by the deconvolution model. **b₁, b₂** pp-ResNet output is deconvolved into two components. **c** The deconvolution model supports maximally 3 components per initial signal. **d** Two peak picking markers are merged by the deconvolution model. **e** Peak picking output deconvolved into three components.

mainly due to displacements of terminal secondary structure elements (e.g., a tilted α-helix near a chain terminus), or inaccurate loop conformations (e.g., more flexible than in the PDB deposition). We investigated the origin of these discrepancies.

2KCD is a 120-residue (14.4 kDa) protein from *Staphylococcus saprophyticus* with an α-β roll architecture. Its dataset comprises 19 spectra (8 backbone, 6 side-chain, and 5 NOESY). The ARTINA structure has a backbone RMSD to PDB reference of 3.13 Å, which is caused by the displacement of the C-terminal α-helix (residues 105–109; Supplementary Fig. 4a). Excluding this 5-residue fragment decreases the RMSD to 2.40 Å (Supplementary Table 11). The

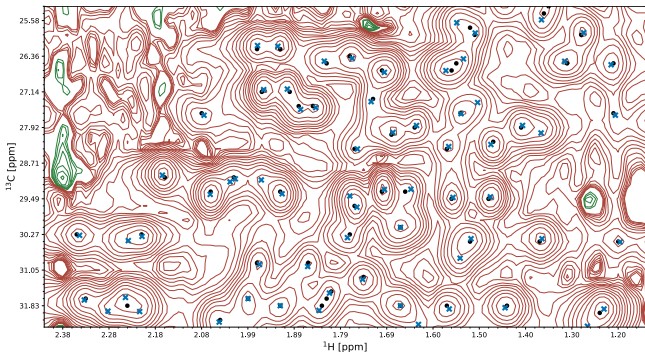

**Fig. 7 | Performance of the peak picking model on a spectrum fragment with high peak overlap.** A fragment of the $^{13}$C-HSQC spectrum of protein 2K0M is shown. Initial signal positions identified by the peak picking model pp-ResNet (black dots) are deconvolved by deconv-ResNet, yielding the final coordinates used for automated assignment and structure determination (blue crosses).

positioning of this helix appears to be uncertain, since an ARTINA calculation without the 4D CC-NOESY spectrum yields a significantly lower RMSD of 1.77 Å (Supplementary Table 12).

2L82 is a de novo designed protein of 162 residues (19.7 kDa) with an αβ 3-layer (αβα) sandwich architecture. Although only 9 spectra (4 backbone, 2 side-chain and 3 NOESY) are available, ARTINA correctly assigned 97.87% backbone and 81.05% side-chain chemical shifts. The primary reason for the high RMSD value of 3.55 Å is again a displacement of the C-terminal α-helix (residues 138–153). The remainder of the protein matches closely the PDB deposition (1.04 Å RMSD, Supplementary Fig. 4b).

The protein with highest RMSD to reference (4.72 Å) in our benchmark dataset is 2M47, a 163-residue (18.8 kDa) protein from *Corynebacterium glutamicum* with an α-β 2-layer sandwich architecture, for which 17 spectra (7 backbone, 7 side chain and 3 NOESY) are available. The main source of discrepancy are two α-helices spanning residues 111–157 near the C-terminus. Nevertheless, the residues contributing to the high RMSD value are distributed more extensively than in 2L82 and 2KCD just discussed. Interestingly, 2 of the 10 structure proposals calculated by ARTINA have an RMSD to reference below 2 Å (1.66 Å and 1.97 Å). In the final structure selection step, our GBT model selected the 4.72 Å RMSD structure as the first choice and 1.66 Å as the second one (Supplementary Fig. 4c). Such results imply that the automated structure determination of this protein is unstable. Since ARTINA returns the two structures selected by GBT with the highest confidence, the user can, in principle, choose the better structure based on contextual information.

In addition to these three case studies, we performed a quantitative analysis of all regular secondary structure elements and flexible loops present in our 100-protein benchmark in order to assess their impact on the backbone RMSD to reference (Supplementary Table 11). All residues in the structurally well-defined regions determined by CYRANGE[29] were assigned to 6 partially overlapping sets: (a) first secondary structure element, (b) last secondary structure element, (c) α-helices, (d) β-sheets, (e) α-helices and β-sheets, and (f) loops. Then, the RMSD to reference was calculated 6 times, each time with one set excluded. In total, for 66 of the 100 proteins the lowest RMSD was obtained if set (f) was excluded from RMSD calculation, and 13% benefited most from removal of the first or last secondary structure element (a or b). Moreover, for 18 out of the 19 proteins with more than 0.5 Å RMSD decrease compared to the RMSD for all well-defined residues, (a), (b), or (f) was the primary source of discrepancy. These results are consistent with our earlier statement that deviations in automatically determined protein structures are mainly caused by terminal secondary structure elements or inaccurate loop conformations.

**Table 1 | Quality of assignments and structures in each refinement cycle**

|  |  | Quantity | Refinement cycle | | |
|---|---|---|---|---|---|
|  |  |  | 1 | 2 | 3 |
| Chemical shift assignment | Initial | Backbone assignment accuracy [%] | 96.12 | 96.19 | **96.34** |
|  |  | Side-chain assignment accuracy [%] | 84.90 | 86.83 | **86.95** |
|  |  | All-atom assignment accuracy [%] | 89.20 | 90.51 | **90.79** |
|  | Refined | Backbone assignment accuracy [%] | 96.78 | 96.92 | **97.22** |
|  |  | Side-chain assignment accuracy [%] | 86.02 | 87.75 | **88.04** |
|  |  | All-atom assignment accuracy [%] | 90.17 | 91.31 | **91.36** |
| Structure calculation |  | CYANA target function value [Å$^2$] | 4.53 | 4.61 | **4.03** |
|  |  | Backbone RMSD to reference [Å] | 1.56 | 1.52 | **1.44** |
|  |  | Heavy-atom RMSD to reference [Å] | 2.21 | 2.07 | **2.02** |
|  |  | Proteins with backbone RMSD to reference ≤ 1 Å [%] | 17 | 20 | 26 |
|  |  | Proteins with backbone RMSD to reference 1–2 Å [%] | 51 | 54 | 51 |
|  |  | Proteins with backbone RMSD to reference 2–3 Å [%] | 18 | 17 | 20 |
|  |  | Proteins with backbone RMSD to reference > 3 Å [%] | 14 | 9 | 3 |

Reported quantities (except for the RMSD distribution in the 4 bottom rows) are median values over the 100 proteins in the benchmark data set. The best metric value in each row is presented in bold. A refinement cycle is a single ARTINA iteration, composed of one execution of the chemical shift refinement cycle (comprising two FLYA[12] executions; rows initial and refined), and the structure refinement cycle (comprising 10 CYANA runs[22]).

## Ablation studies

During the experiment, we captured the state of each structure determination at 9 time-points, 3 per structure determination cycle: (a) after the initial FLYA shift assignment, (b) after GNN shift refinement, and (c) after structure calculation (Fig. 1). Comparative analysis of these states allowed us to quantify the contribution of different ARTINA components to the structure determination process (Table 1).

The results show a strong benefit of the refinement cycles, as quantities reported in Table 1 consistently improve from cycle 1 to 3. The majority of benchmark proteins converge to the correct fold after the first cycle (1.56 Å median backbone RMSD to reference), which is further refined to 1.52 Å in cycle 2 and 1.44 Å in cycle 3. Additionally, within each chemical shift refinement cycle, improvements in assignment accuracy resulting from the GNN predictions are observed. This quantity also increases consistently across all refinement cycles, in particular for side-chains. Refinement cycles are particularly advantageous for large and challenging systems, such as 2LF2, 2M7U, or 2B3W, which benefit substantially in cycles 2 and 3 from the presence of the approximate protein fold in the chemical shift assignment step.

## Impact of 4D NOESY experiments

As presented in Fig. 2, 26 out of 100 benchmark datasets contain 4D CC-NOESY spectra, which require long measurement times and were used in the manual structure determination. To quantify their impact, we performed automated structure determinations of these 26 proteins with and without the 4D CC-NOESY spectra (Supplementary Table 12).

On average, the presence of 4D CC-NOESY improves the backbone RMSD to reference by 0.15 Å (decrease from 1.88 to 1.73 Å) and has less than 1% impact on chemical shift assignment accuracy. However, the impact is non-uniform. For three proteins, 2KIW, 2L8V, and 2LF2, use of the 4D CC-NOESY decreased the RMSD by more than 1 Å. On the other hand, there is also one protein, 2KCD, for which the RMSD decreased by more than 1 Å by *excluding* the 4D CC-NOESY.

These results suggest that overall the amount of information stored in 2D/3D experiments is sufficient for ARTINA to reach close to optimal performance, and only modest improvement can be achieved by introducing additional information redundancy from 4D CC-NOESY spectra.

## Automated chemical shift assignment

Apart from structure determination, our data analysis pipeline for protein NMR spectroscopy can address an array of problems that are nowadays approached manually or semi-manually. For instance, ARTINA can be stopped after visual spectrum analysis, returning positions and intensities of cross-peaks that can be utilized for any downstream task, not necessarily related to protein structure determination.

Alternatively, a single chemical shift refinement cycle can be performed to get automatically assigned cross-peaks from spectra and sequence. We evaluated this approach with three sets of spectra: (i) Exclusively backbone assignment spectra were used to assign N, C$^\alpha$, C$^\beta$, C', and H$^N$ shifts. With this input, ARTINA assigned 92.40% (median value) of the backbone shifts correctly. (ii) All through-bond but no NOESY spectra were used to assign the backbone and side-chain shifts. This raised the percentage of correct backbone assignments to 94.20%. (iii) The full data set including NOESY yielded 96.60% correct assignments of the backbone shifts. These three experiments were performed for the 45 benchmark proteins, for which CBCANH and CBCAcoNH, as well as either HNCA and HNcoCA or HNCO and HNcaCO experiments were available. The availability of NOESY spectra had a large impact on the side-chain assignments: 86.00% were correct for the full spectra set iii, compared to 73.70% in the absence of NOESY spectra (spectra set ii). The presence of NOESY spectra consistently improved the chemical shift assignment accuracy of all amino acid types (Supplementary Tables 13 and 14). The improvement is particularly strong for aromatic residues (Phe, 61.6 to 76.5%, Trp 52.5 to 80%, and Tyr 71.4 to 89.7%), but not limited to this group.

## Discussion

The results obtained with ARTINA differ in several aspects substantially from previous approaches towards automating protein NMR analysis[3,4,7,12,17–19,35]. First, ARTINA comprehends the entire workflow from spectra to structures rather than individual steps in it, and there are strictly no manual interventions or protein-specific parameters to be adapted. Second, the quality of the results regarding peak identification, resonance assignments, and structures have been assessed on a large and diverse set of 100 proteins; for the vast majority of which they are on par with what can be achieved by human experts. Third, the method provides a two-orders-of-magnitude leap in efficiency by providing assignments and a structure within hours of computation time rather than weeks or months of human work. This reduces the effort for a protein structure determination by NMR essentially to the preparation of the sample and the measurement of the spectra. Its implementation in the https://nmrtist.org webserver (Supplementary

Movie 5) encapsulates its complexity, eliminates any intermediate data and format conversions by the user, and enables the use of different types of high-performance hardware as appropriate for each of the subtasks. ARTINA is not limited to structure determination but can be used equally well for peak picking and resonance assignment in NMR studies that do not aim at a structure, such as investigations of ligand binding or dynamics.

Although ARTINA has no parameters to be optimized by the user, care should be given to the preparation of the input data, i.e., the choice, measurement, processing, and specification of the spectra. Spectrum type, axes, and isotope labeling declarations must be correct, and chemical shift referencing consistent over the entire set of spectra. Slight variations of corresponding chemical shifts within the tolerances of 0.03 ppm for $^1$H and 0.4 ppm for $^{13}$C/$^{15}$N can be accommodated, but larger deviations, resulting, for instance, from the use of multiple samples, pH changes, protein degradation, or inaccurate referencing, can be detrimental. Where appropriate, ARTINA proposes corrections of chemical shift referencing[36]. Furthermore, based on the large training data set, which comprises a large variety of spectral artifacts, ARTINA largely avoids misinterpreting artifacts as signals. However, with decreasing spectral quality, ARTINA, like a human expert, will progressively miss real signals.

Regarding protein size and spectrum quality, limitations of ARTINA are similar to those encountered by a trained spectroscopist. Machine-learning-based visual analysis of spectra requires signals to be present and distinguishable in the spectra. ARTINA does not suffer from accidental oversight that may affect human spectra analysis. On the other hand, human experts may exploit contextual information to which the automated system currently has no access because it identifies individual signals by looking at relatively small, local excerpts of spectra.

In this paper, we used all spectra that are available from the earlier manual structure determination. For most of the 100 proteins, the spectra data set has significant redundancy regarding information for the resonance assignment. Our results indicate that one can expect to obtain good assignments and structures also from smaller sets of spectra[37], with concomitant savings of NMR measurement time. We plan to investigate this in a future study.

The present version of ARTINA can be enhanced in several directions. Besides improving individual models and algorithms, it is conceivable to integrate the so far independently trained collection of machine learning models, plus additional models that replace conventional algorithms, into a coherent system that is trained as a whole. Furthermore, the reliability of machine learning approaches depends strongly on the quantity and quality of training data available. While the collection of the present training data set for ARTINA was cumbersome, from now on it can be expected to expand continuously through the use of the https://nmrtist.org website, both quantitatively and qualitatively with regard to greater variability in terms of protein types. spectral quality, source laboratory, data processing (including non-linear sampling), etc., which can be exploited in retraining the models. ARTINA can also be extended to use additional experimental input data, e.g., known partial assignments, stereospecific assignments, $^3J$ couplings, residual dipolar couplings, paramagnetic data, and H-bonds. Structural information, e.g., from AlphaFold[38], can be used in combination with reduced sets of NMR spectra for rapid structure-based assignment. Finally, the range of application of ARTINA can be generalized to small molecule-protein complexes relevant for structure-activity relationship studies in drug research, protein-protein complexes, RNA, solid state, and in-cell NMR.

Overall, ARTINA stands for a paradigm change in biomolecular NMR from a time-consuming technique for specialists to a fast method open to researchers in molecular biology and medicinal chemistry. At the same time, in a larger perspective, the appearance of generally highly accurate structure predictions by AlphaFold[38] is revolutionizing

structural biology. Nevertheless, there remains space for the experimental methods, for instance, to elucidate various states of proteins under different conditions or in dynamic exchange, or for studying protein-ligand interaction. Regarding ARTINA, one should keep in mind that its applications extend far beyond structure determination. It will accelerate virtually any biological NMR studies that require the analysis of multidimensional NMR spectra and chemical shift assignments. Protein structure determination is just one possible ARTINA application, which is both demanding in terms of the amount and quality of required experimental data and amenable to quantitative evaluation.

## Methods

### Spectrum benchmark collection
To collect the benchmark of NMR spectra (Fig. 2 and Supplementary Table 2), we implemented a crawler software, which systematically scanned the FTP server of the BMRB data bank[39], identifying data files relevant to our study. Additional datasets were obtained by setting up a website for the deposition of published data (https://nmrdb.ethz.ch), from our collaboration network, or had been acquired internally in our laboratory. NMR data was collected from these channels either in the form of processed spectra (Sparky[40], NMRpipe[41], XEASY[42], Bruker formats), or in the form of time-domain data accompanied by depositor-supplied NMRpipe processing scripts. No additional spectra processing (e.g., baseline correction) was performed as part of this study.

The most challenging aspects of the benchmark collection process were: scarcity of data—only a small fraction of all BMRB depositions are accompanied by uploaded spectra (or time-domain data), lack of standards for NMR data depositions—each protein data set had to be prepared manually, as the original data was stored in different formats (spectra name conventions, axis label standards, spectra data format), and difficulties in correlating data files deposited in the BMRB FTP site with contextual information about the spectrum and the sample (e.g., sample characteristics, measurement conditions, instrument used). Manually prepared (mostly NOESY) peak lists, which are available from the BMRB for some of the proteins in the benchmark, were not used for this study.

Different approaches to 3D $^{13}$C-NOESY spectra measurement had to be taken into account: (i) Two separate $^{13}$C NOESY for aliphatic and aromatic signals. These were analyzed by ARTINA without any special treatment. We used ALI, ARO tags (Supplementary Movie S5) to provide the information that only either aliphatic or aromatics shifts are expected in a given spectrum. (ii) Simultaneous NC-NOESY. These spectra were processed twice to have proper scaling of the $^{13}$C and $^{15}$N axes in ppm units, and cropped to extract $^{15}$N-NOESY and $^{13}$C-NOESY spectra. If nitrogen and carbon cross-peak amplitudes have different signs, we used POS, NEG tags to provide the information that only either positive or negative signals should be analyzed. (iii) Aliphatic and aromatic signals in a single $^{13}$C-NOESY spectrum. These measurements do not require any special treatment, but proper cross-peak unfolding plays a vital role in aromatic signals analysis.

### Overview of the ARTINA algorithm
ARTINA uses as input only the protein sequence and a set of NMR spectra, which may contain any combination of 25 experiments currently supported by the method (Supplementary Table 1). Within 4–20 h of computation time (depending on protein size, number of spectra, and computing hardware load), ARTINA determines: (a) cross-peak positions for each spectrum, (b) chemical shift assignments, (c) distance restraints from NOESY spectra, and (d) the protein structure. The whole process does not require any human involvement, allowing rapid protein NMR assignment and structure determination by non-experts.

The ARTINA workflow starts with *visual spectrum analysis* (Fig. 1), wherein cross-peak positions are identified in frequency-domain NMR spectra using deep residual neural networks (ResNet)[24]. Coordinates of signals in the spectra are passed as input to the FLYA automated assignment algorithm[12], yielding initial chemical *shift assignments*. In the subsequent chemical *shift refinement* step, we bring to the workflow contextual information about thousands of protein structures solved by NMR in the past using a deep GNN[25] that was trained on BMRB/PDB depositions. Its goal is to predict expected values of yet missing chemical shifts, given the shifts that have already been confidently and unambiguously assigned by FLYA. With these GNN predictions as additional input, the cross-peak positions are reassessed in a second FLYA call, which completes the *chemical shift refinement cycle* (Fig. 1).

In the *structure refinement cycle*, 10 variants of NOESY peak lists are generated, which differ in the number of cross-peaks selected from the output of the visual spectrum analysis by varying the confidence threshold of a signal selected by ResNet between 0.05 and 0.5. Each set of NOESY peak lists is used in an independent CYANA structure calculation[22,23], yielding *10 intermediate structure proposals* (Fig. 1). The structure proposals are ranked in the *intermediate structure selection* step based on 96 features with a dedicated GBT model. The selected best structure proposal is used as contextual information in a consecutive FLYA run, which closes the *structure refinement cycle*.

After the two initial steps of visual spectrum analysis and initial chemical shift assignment, ARTINA interchangeably executes refinement cycles. The chemical shift refinement cycle provides FLYA with tighter restraints on expected chemical shifts, which helps to assign ambiguous cross-peaks. The structure refinement cycle provides information about possible through-space contacts, allowing identified cross-peaks (especially in NOESY) to be reassigned. The high-level concept behind the interchangeable execution of refinement cycles is to iteratively update the protein structure given fixed chemical shifts, and update chemical shifts given the fixed protein structure. Both refinement cycles are executed three times.

## Automated visual analysis of the spectrum

We established two machine learning models for the visual analysis of multidimensional NMR spectra (see downloads in the Code availability section). In their design, we made no assumptions about the downstream task and the 2D/3D/4D experiment type. Therefore, the proposed models can be used as the starting point of our automated structure determination procedure, as well as for any other task that requires cross-peak coordinates.

The automated visual analysis starts by selecting all extrema $\boldsymbol{x} = \{\boldsymbol{x}_1, \boldsymbol{x}_2, \ldots, \boldsymbol{x}_N\}$, $\boldsymbol{x}_n \in \mathbb{N}^D$ in the NMR spectrum, which is represented as a $D$-dimensional regular grid storing signal intensities at discrete frequencies. We formulated the peak picking task as an object detection problem, where possible object positions are confined to $\boldsymbol{x}$. This task was addressed by training a deep residual neural network[24], in the following denoted as peak picking ResNet (pp-ResNet), which learns a mapping $\boldsymbol{x}_n \rightarrow [0, 1]$ that assigns to each signal extremum a real-valued score, which resembles its probability of being a true signal rather than an artefact.

Our network architecture is strongly linked to ResNet-18[24]. It contains 8 residual blocks, followed by a single fully connected layer with sigmoidal activation. After weight initialization with Glorot Uniform[43], the architecture was trained by optimizing a binary cross-entropy loss using Adam[44] with learning rate $10^{-4}$ and gradient clipping of 0.5.

To establish an experimental training dataset for pp-ResNet, we normalized the 1329 spectra in our benchmark with respect to resolution (adjusting the number of data grid points per unit chemical shift (ppm) using linear interpolation) and signal amplitude (scaling the spectrum by a constant). Subsequently, 675,423 diverse 2D fragments

of size $256 \times 32 \times 1$ were extracted from the normalized spectra $\boldsymbol{x}$ and manually annotated, yielding 98,730 positive and 576,693 negative class training examples. During the training process, we additionally augmented this dataset by flipping spectrum fragments along the second dimension (32 pixels), stretching them by 0–30% in the first and second dimensions, and perturbing signal intensities with Gaussian noise addition.

The role of the pp-ResNet is to quickly iterate over signal extrema in the spectrum, filtering out artefacts and selecting approximate cross-peak positions for the downstream task. The relatively small network architecture (8 residual blocks) and input size of 2D $256 \times 32$ image patches make it possible to analyze large 3D $^{13}$C-resolved NOESY spectra in less than 5 min on a high-end desktop computer. Simultaneously, the first dimension of the image patch (256 pixels) provides long-range contextual information on the possible presence of signals aligned with the current extremum (e.g., $C^\alpha$, $C^\beta$ cross-peaks in an HNCACB spectrum).

Extrema classified with high confidence as true signals by pp-ResNet undergo subsequent analysis with a second deep residual neural network (deconv-ResNet). Its objective is to perform signal deconvolution, based on a 3D spectrum fragment ($64 \times 32 \times 5$ voxels) that is cropped around a signal extremum selected by pp-ResNet. This task is defined as a regression problem, where deconv-ResNet outputs a $3 \times 3$ matrix storing 3D coordinates of up to 3 deconvolved peak components, relative to the center of the input image. To ensure permutation invariance with respect to the ordering of components in the output coordinate matrix, and to allow for a variable number of 1–3 peak components, the architecture was trained with a Chamfer distance loss[45].

Since deconv-ResNet deals only with true signals and their local neighborhood, its training dataset can be conveniently generated. We established a spectrum fragment generator, based on rules reflecting the physics of NMR, which produced 110,000 synthetic training examples (Supplementary Fig. 1) having variable (a) numbers of components to deconvolve (1–3), (b) signal-to-noise ratio, (c) component shapes (Gaussian, Lorentzian, and mixed), (d) component amplitude ratios, (e) component separation, and (f) component neighborhood type (i.e., NOESY-like signal strips or HSQC-like 2D signal clusters). The deconv-ResNet model was thus trained on fully synthetic data.

## Signal unaliasing

To use ResNet predictions in automated chemical shift assignment and structure calculation, detected cross-peak coordinates must be transformed from the spectrum coordinate system to their true resonance frequencies. We addressed the problem of automated signal unfolding with the classical machine learning approach to density estimation.

At first, we generated $10^5$ cross-peaks associated with each experiment type supported by ARTINA (Supplementary Table 1). In this process, we used randomly selected chemical shift lists deposited in the BMRB database, excluding depositions associated with our benchmark proteins. Subsequently, we trained a Kernel Density Estimator (KDE):

$$p_e(\boldsymbol{x}) = \frac{1}{N_e} \sum_{i=1}^{N_e} \kappa(\boldsymbol{x} - \boldsymbol{x}_i^{(e)}) \tag{1}$$

which captures the distribution $p_e(\boldsymbol{x})$ of true peaks being present at position $\boldsymbol{x}$ in spectrum type $e$, based on $N_e = 10^5$ cross-peaks coordinates $\boldsymbol{x}_i^{(e)}$ generated with BMRB data, and $\kappa$ being the Gaussian kernel.

Unfolding a $k$-dimensional spectrum is defined as a discrete optimization problem, solved independently for each cross-peak $\boldsymbol{x}_j^{(e)}$

observed in a spectrum of type $e$:

$$s^* = \arg \max_s p_e(x_j^{(e)} + w \circ s) \qquad (2)$$

where $w \in \mathbb{R}^k$ is a vector storing the spectral widths in each dimension (ppm units), $\circ$ is element-wise multiplication, $s \in \mathbb{Z}^k$ is a vector indicating how many times the cross-peak is unfolded in each dimension, and $s^* \in \mathbb{Z}^k$ is the optimal cross-peak unfolding.

As long as regular and folded signals do not overlap or have different signs in the spectrum, KDE can unfold the peak list regardless of spectrum dimensionality. The spectrum must not be cropped in the folded dimension, i.e., the folding sweep width must equal the width of the spectrum in the corresponding dimension.

All 2D/3D spectra in our benchmark were folded in at most one dimension and satisfy the aforementioned requirements. However, the 4D CC-NOESY spectra satisfy neither, as regular and folded peaks both overlap and have the same signal amplitude sign. This introduces ambiguity in the spectrum unfolding that prevents direct use of the KDE technique. To retrieve original signal positions, 4D CC-NOESY cross-peaks were unfolded to overlap with signals detected in 3D $^{13}$C-NOESY. In consequence, 4D CC-NOESY unfolding depended on other experiments, and individual 4D cross-peaks were retained only if they were confirmed in a 3D experiment.

## Chemical shift assignment

Chemical shift assignment is performed with the existing FLYA algorithm[12] that uses a genetic algorithm combined with local optimization to find an optimal matching between expected and observed peaks. FLYA uses as input the protein sequence, lists of peak positions from the available spectra, chemical shift statistics, either from the BMRB[39] or the GNN described in the next section, and, if available, the structure from the previous refinement cycle. The tolerance for the matching of peak positions and chemical shifts was set to 0.03 ppm for $^1$H, and 0.4 ppm for $^{13}$C/$^{15}$N shifts. Each FLYA execution comprises 20 independent runs with identical input data that differ in the random numbers used in the optimization algorithm. Nuclei for which at least 80% of the 20 runs yield, within tolerance, the same chemical shift value are classified as reliably assigned[12] and used as input for the following chemical shift refinement step.

## Chemical shift refinement

We used a graph data structure to combine FLYA-assigned shifts with information from previously assigned proteins (BMRB records) and possible spatial interactions. Each node corresponds to an atom in the protein sequence, and is represented by a feature vector composed of (a) a one-hot encoded atom type code (e.g., $C^\alpha$, $H^\beta$), (b) a one-hot encoded amino acid type, (c) the value of the chemical shift assigned by FLYA (only if a confident assignment is available, zero otherwise), (d) atom-specific BMRB shift statistics (mean and standard deviation), and (e) 30 chemical shift values obtained from BMRB database fragments. The latter feature is obtained by searching BMRB records for assigned 2–3-residue fragments that match the local protein sequence and have minimal mean-squared-error (MSE) to shifts confidently assigned by FLYA (non-zero values of feature (c) in the local neighborhood of the atom). The edges of the graph correspond to chemical bonds or skip connections. The latter connect the $C^\beta$ atom of a given residue with $C^\beta$ atoms 2, 3, and 5 residues apart in the amino acid sequence, and have the purpose to capture possible through-space influence on the chemical shift that is typically observed in secondary structure elements.

The chemical shift refinement task is defined as a node regression problem, where an expected value of the chemical shift is predicted for each atom that lacks a confident FLYA assignment. This task is addressed with a DeepGCN model[25,26] that was trained on 28,400 graphs extracted from 2840 referenced BMRB records[39]. Each training example was created by building a fully assigned graph out of a single BMRB record, and dropping chemical shift values (feature (c) above) for randomly chosen atoms that FLYA typically assigns either with low confidence or inaccurately.

Our DeepGCN model is designed specifically for de novo structure determination, as it uses only the protein sequence and partial shift assignments to estimate values of missing chemical shifts. Its predictions are used to guide the FLYA genetic algorithm optimization[12] by reducing its search range for assignments. The precise final chemical shift value is always determined by the position of a signal in the spectrum, rather than the model prediction alone.

## Torsion angle restraints

Before each structure calculation step, torsion angle restraints for the $\phi$ and $\psi$ angles of the polypeptide backbone were obtained from the current backbone chemical shifts using the program TALOS-N[21]. Restraints were only generated if TALOS-N classified the prediction as 'Good', 'Strong', or 'Generous'. Given a TALOS-N torsion angle prediction of $\phi \pm \Delta\phi$, the allowed range of the torsion angle was set to $\phi \pm \max(\Delta\phi, 10°)$ for 'Good' and 'Strong' predictions, and $\phi \pm 1.5 \max(\Delta\phi, 10°)$ for 'Generous' predictions, and likewise for $\psi$.

## Structure calculation and selection

Given the chemical shift assignments and NOESY cross-peak positions and intensities, the structure is calculated with CYANA[23] using the established method[22] that comprises 7 cycles of NOESY cross-peak assignment and structure calculation, followed by a final structure calculation. In total, $8 \times 100$ conformers are calculated for a given input data set using 30,000 torsion angle dynamics steps per conformer. The 20 conformers with the lowest final target function value are chosen to represent the solution structure proposal. The entire combined NOESY assignment and structure calculation procedure is executed independently 10 times based on 10 variants of NOESY peak lists, which differ in the number of cross-peaks selected from the output of the visual spectrum analysis. The first set generously includes all signals selected by ResNet with confidence ≥0.05. The other variants of NOESY peak lists follow the same principle with increasingly restrictive confidence thresholds of 0.1, 0.15, …, 0.5.

The CYANA structures calculations are followed by a structure selection step, wherein the 10 intermediate structure proposals are compared pairwise by a Gradient Boosted Tree (GBT) model that uses 96 features from each structure proposal (including the CYANA target function value[23], number of long-range distance restraints, etc.; for details, see downloads in the Code availability section) to rank the structures by their expected accuracy. The best structure from the ranking is subsequently used as contextual information for the chemical shift refinement cycle (Fig. 1), or returned as the final outcome of ARTINA. The second-best final structure is also returned for comparison.

To train GBT, we collected a set of successful and unsuccessful structure calculations with CYANA. Each training example was a tuple $(s_i, r_i)$, where $s_i$ is the vector of features extracted from the CYANA structure calculation output, and $r_i$ is the RMSD of the output structure to the PDB reference. The GBT was trained to take the features $s_i$ and $s_j$ of two structure calculations with CYANA as input, and to predict a binary order variable $o_{ij}$, such that $o_{ij} = 1$ if $r_i < r_j$, and 0 otherwise. Importantly, the deposited PDB reference structures were not used directly in the GBT model training (they are used only to calculate the RMSDs). Consequently, the GBT model is unaffected by methodology and technicalities related to PDB deposition (e.g., the structure calculation software used to calculate the deposited reference structure).

## Structure accuracy estimate

As an accuracy estimate for the final ARTINA structure, a predicted RMSD to reference (pRMSD) is calculated from the ARTINA results

(without knowledge of the reference PDB structure). It aims at reproducing the actual RMSD to reference, which is the RMSD between the mean coordinates of the ARTINA structure bundle and the mean coordinates of the corresponding reference PDB structure bundle for the backbone atoms N, $C^\alpha$, C' in the residue ranges as given in Supplementary Table 4. The predicted RMSD is given by pRMSD = $(1 - t) \times$ 4 Å, where, in analogy to the GDT_HA value[46], $t$ is the average fraction of the RMSDs ≤ 0.5, 1, 2, 4 Å between the mean coordinates of the best ARTINA candidate structure bundle and the mean coordinates of the structure bundles of the 9 other structure proposals. Since $t \in [0, 1]$, the pRMSD is always in the range of 0–4 Å, grouping all "bad" structures with expected RMSD to reference ≥ 4 Å at pRMSD = 4 Å.

## Reporting summary

Further information on research design is available in the Nature Research Reporting Summary linked to this article.

## Data availability

References structures: PDB Protein Data Bank (https://www.rcsb.org/; accession codes in Fig. 2 and Supplementary Table 3).

Spectra and reference assignments: BMRB Biological Magnetic Resonance Data Bank (https://bmrb.io/; entry IDs in Supplementary Table 3).

Peak lists, assignments, and structures: https://nmrtist.org/static/public/publications/artina/ARTINA_results.zip and in the ETH Research Collection under DOI 10.3929/ethz-b-000568621.

Source data for Figs. 2, 4, and 5 is available in Supplementary Tables 2, 4, and 5, respectively.

## Code availability

The ARTINA algorithm is available as a webserver at https://nmrtist.org. pp-ResNet, deconv-ResNet, GNN, and GBT are available for download in binary form, together with architecture schemes, example input data, model input description, and source code that allows to read model files and make predictions (https://github.com/PiotrKlukowski/ARTINA, https://nmrtist.org/static/public/publications/artina/models/ {ARTINA_peak_picking.zip, ARTINA_peak_deconvolution.zip, ARTINA_ shift_prediction.zip, ARTINA_structure_ranking.zip}). These files provide a full technical specification of the components developed within ARTINA, and allow for their independent use in Python.

Existing software used: Python (https://www.python.org/), CYANA (https://www.las.jp/), TALOS-N (https://spin.niddk.nih.gov/bax/software/TALOS-N).

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

## Acknowledgements

We thank Drs. Frédéric Allain, Fred Damberger, Hideo Iwai, Harindranath Kadavath, Julien Orts, and Dean Strotz for providing unpublished spectra. This project has received funding from the European Union's Horizon 2020 research and innovation program under the Marie Sklodowska-Curie grant agreement No 891690 (P.K.), and a Grant-in-Aid for Scientific Research of the Japan Society for the Promotion of Science (P.G., 20 K06508).

## Author contributions

P.K. prepared training and test data sets, designed and trained machine learning models, performed experiments described in the manuscript, and implemented ARTINA within the nmrtist.org web platform. P.K. and P.G. wrote the software. P.K., R.R., and P.G. conceived the project, analyzed the results, and wrote the manuscript.

## Competing interests

The authors declare no competing interests.
