## [Peer Review File · Nature Communications]

Rapid protein assignments and structures from raw NMR spectra with the deep learning technique ARTINAReviewer #1 (Remarks to the Author):

The manuscript describes the AI-based automatic determination of protein structures of proteins with a molecular weight of up to 20kDa in a completely automated manner without human expert interference between recording of NMR spectra to structure. This is a truly remarkable achievement, unmet by previous extensive efforts to solve this most outstanding NMR protein structure determination problem. As such, the manuscript is of highest interest and quality so that it should be published essentially as is. There are two minor comments, one of which I would really like to see conducted, the second is more stylistic.

Minor comments:

1.) The authors should not only provide the pdb codes for the proteins used but also their name and function and also where structures have been published for these proteins, if the structures have been published and not simply deposited.

2.) Stylistic comment: On page 4, the authors write: "...to reproduce the ground truth protein structures".

With a tinkle in the eyes, I confessed that I do not know what a ground truth structure is. Maybe they want to say: bona fide correct structure?

Reviewer #2 (Remarks to the Author):

The work presents a machine-learning based approach (named ARTINA) to automatically determine protein structure from processed NMR spectra. The method is built around a new peak-picking procedure and two already existing methods for automated chemical shift assignment (FLYA) and structure calculation (CYANA). The main originalities of the approach are i) the use of deep residual neural networks for NMR peaks identification in multi-dimensional spectra and deconvolution and ii) a fully integrated pipeline, with several cycles of assignments and structure calculation with FLYA and CYANA. For training of the neural networks and validation of the performances of the approach, a benchmark dataset was constructed, that includes 1329 processed NMR spectra from 100 proteins, along with the corresponding chemical shifts (BMRB) and atomic coordinates (PDB) records. The two main criteria chosen by the authors to evaluate the efficacy of ARTINA are i) the percentage of correctly assigned chemical shifts when compared to BMRB records and ii) the closeness of structure solutions from CYANA to the PDB records (measured as a RMSD). Overall, ARTINA correctly assigns more than 90% of the chemical shifts and the median backbone RMSD of the final structures with the reference coordinates is below 1.5 Å, across the whole benchmark set. Additionally, the authors evaluated the impact of the recycling between FLYA and CYANA and also the usefulness of including 4D NOESY spectra.

First, I would like to congratulate the authors on having tackled (and succeeded) in establishing such a benchmark dataset that was definitely missing in the field (although there were some attempts from the CASD initiative and a substantial amount of spectra already available from BMRB as time-domain data, mainly deposited by structural genomics consortia).

The manuscript is sound and well written. The webserver implementation of ARTINA is functional (tested on one dataset provided by the authors on the nmrtist.org website). Knowing the excellent track record of the Riek and Güntert groups in this field, I have no doubt that ARTINA will make the life of many NMR spectroscopists much easier.

As an NMR spectroscopist myself, I'm convinced that this work represents a great advance in the field of protein NMR. But, with the new developments of artificial intelligence algorithms we are currently witnessing in biological science, and especially the recent breakthroughs from DeepMind's AlphaFold or David Baker's RosettaFold in structure prediction, I (provocatively) will question the relevance of an end-to-end NMR

analysis pipeline (despite being apparently the best one) ? Indeed, it has recently been shown that AlphaFold can generate models with similar or even better fit to NMR data than for the experimental NMR or X-ray structures (Terejo et al, 2022 <https://doi.org/10.1101/2022.03.09.483701> and Huang et al., 2022 <https://doi.org/10.1002/prot.26246>). Maybe the future of NMR lies more in dynamics analysis, ligand screening, interaction mapping rather than in structure determination ? The authors must discuss more the real advances of their method in this new context of structural biology, a.k.a. the post-AlphaFold era.

However, before considering the manuscript for possible publication, I have the following main concerns (that I will detail more below):

- the incomplete methodological details
- the lack of comparison of the new peak-picker (which lies at the core of the ARTINA workflow) to other existing software with the same purpose
- the possible biases in training and inference
- the lack of a "reliability" score for structure proposals and most importantly:
- the unavailability of source code and trained models from the deep-learning parts

1) The description of the methods is not complete, i.e. not self-sufficient to reproduce the results (or re-implement the presented algorithm):

Schematics of network architectures for pp-ResNet, deconv-ResNet and the DeepGNN used for expected chemical shifts must be presented since their description is lacking some important details.

The authors state that a 5-fold cross validation was applied to assess the performance (using only 80% for training and 20 % for testing). What are thus the RMSD values reported in Supp Table 3, Table 1 and Figure 3 and 4 for each entry? Is this the average over the 5 tests ? In the current implementation available on the server, which training models for pp-ResNet and GBT are used ? Was it trained on 100% of the data set ?

How is calculated the RMSD between structure solutions from ARTINA and reference PDB entries ? Since both reference and query are ensembles of conformation, and a single value is reported for the RMSD to reference, one can question if it is "average vs average of each ensemble" or "average of pairwise RMSD between the target ensemble and the entry ensemble" or else... Additionally, in the 2nd round of CASD, a threshold of 1.5 Å was used as criterion to classify a structure solution as accurate when compared to a reference (or ground truth). Here the authors use 3Å to select entries worth being analyzed more or for the count the overall performance. Can the authors justify this 3 Å criterion ?

The list of BMRB entries used should be given in SI (added to Supp. Table 3 for instance) along with the source of the data and the reference structure (paper citation if there is no PDB and/or BMRB entries)

TALOS is briefly mentioned as a dependency at the end of the manuscript but its use is not documented. The details on how TALOS was applied must be given in the methods section.

The GBT (for structure ranking) was trained on the benchmark data set (page 3 "Automated protein structure determination") using 96 features, including CYANA target function and restraints statistics. This part must be presented in a more detailed fashion because in the current state, it cannot be re-implemented solely from the method description. Especially how the training was done. If the ground truth is PDB record, how the features related to distance restraints was obtained? What are the 96 features ? We need to know that to be sure that there are no circular analysis (GBT can select good

structure only from the benchmark ?). This question is also related to my other points about having a possible bias in the benchmark since 80% of the PDB records were originally calculated by CYANA.

2) I am surprised not to see any comparisons between the new peak-picker (pp-Resnet) and other automated peak-picking approaches. The authors should compare the performance (at least for a subset of the benchmark) with some of the popular algorithms available for automated peak-picking among NMRnet, CCPN, NMRViewJ, PICKY, DEEP Picker, CYPICK, AUTOPSY, CV-Peak Picker....

In addition, DEEP Picker, the most recent peak-picker (with deconvolution) also based on deep-learning must be cited since the approach share some similarities with ppResNet and deconv-Resnet (Li et al, 2021 <https://doi.org/10.1038/s41467-021-25496-5>)

3) 80% of protein in the benchmark come from the North East Structural Genomics (NESG) consortium. I suspect this would introduce a bias related to data collection and processing practices? The authors should show their approach perform as good with non-NESG as with NESG entries. Possibly re-train on NESG entries only and predict on the remainder (nonstructural genomics entries and RIKEN and CESG entries)

I counted 67 PDB entries from the data set (~70%) that employed CYANA for the structure calculation of the deposited coordinates. Would that too introduce a bias, possibly in the GBT ranking ? Authors must consider that, evaluate if accuracy is the same in the CYANA and non-CYANA reference structure. Possibly re-train on CYANA and predict on non-CYANA entries ?

5) The authors emphasizes that ARTINA is usable by non-expert. If the authors want to put this argument forward, the software must then return associated confidence values at each output steps: for peak-picking, it's already the case with Resnet; for assignments, FLYA reports some statistics and also strong assignments, but a confidence estimate is missing for structure proposals. If the whole process is fully automated, with no human intervention, it is necessary to guide for the user to choose between the solutions proposals, especially when they are significantly different. I'm surprised the authors didn't use the predicted resolution estimator that they published some years ago (Bagaria et al. *Comput Biol Chem.* 2013).

I thus strongly encourage the authors to include one of the two following metrics to provide a reliability score associated with the structure proposal.

- ANSURR scores as an independent measure of accuracy (Fowler et al., 2020 <https://doi.org/10.1038/s41467-020-20177-1>). I would also be delighted the see ANSURR score computed for reference and for ARTINA solution in the presented benchmark. (RMSD does not tell if a structure is as good or better or worse, just if it's structurally close or not)

- DP-score to estimate how well the solution structures explain the NOESY data (as main source of restraints) (Huang et al. 2015 [doi:10.1007/s10858-015-9955-2](https://doi.org/10.1007/s10858-015-9955-2))

The example of the 2M47, for which the best ranked solution is 4.7 Å from the reference but the 2nd one is much closer, is very interesting. On that point, the authors concluded that automated structure determination is unstable for this protein. But a few years ago, the group of P. Guntert published a new approach name consensus calculation (Buchner L et al. 2015 [doi: 10.1016/j.str.2014.11.014](https://doi.org/10.1016/j.str.2014.11.014).) that was designed exactly to cope with such situations. If the authors decided not to implement the consensus approach as a final step of the pipeline, they should discuss why.

4) ARTINA can be used on a webserver. However, it is not possible to download the software. The code for inference of structure from spectra (end-to-end) must be made available in a repository. If some pieces of software have closed licensing (I think about CYANA for instance), then it would simply become a requirement and it would not be

mandatory to provide it (but rather let to the user to make sure it is installed properly, following the indications of the code repository).

We are now in the era of open science and even Google (a definitely "for profit" company !) made the code and models of its structure prediction AlphaFold available for free with a creative common license. I acknowledge the effort of the authors for making the results from their benchmark available but if other groups wants to build on the advances from ResNet or the GBT for instance, those must be available (at least in binary code, the best would be open source).

Other remarks and/or questions:

- It is said that ARTINA correctly assigned 90.39 % of chemical shifts. Are all assignment or only the ones classified as strong used to assess the accuracy ?
- How would this work on dimeric protein ? Also, I noticed that two entries in the benchmark are in fact protein/RNA complexes. It would be interesting to see the amount of RNA peaks that could not possibly be assigned and how ARTINA did cope with that ? It would also an infesting point to add to the discussion.
- Page 5, 1st paragraph, line 6: "Nevertheless, the residues contributing to the high RMSD value are distributed more extensively than in 2L82 and 2M47 just discussed." 2M47 should be 2KCD instead.
- ARTINA has no free parameters and human intervention is not required. But what if some additional NMR data are available (RDC, J-couplings, H-bond from HDX) or possibly an initial model (possibly from AlphaFold)? How could it integrated in the ARTINA pipeline ?
- Other webservers exist for chemical shift assignment and/or NMR structure calculation . Some of them must be cited (on the top of my head, the ARIAweb and I-Pine webservers).
- For the training of pp-ResNet, manual annotation of 670 000+ 2D spectrum fragments was necessary. To me, this step include some subjectivity (conservative or aggressive annotation?). How does it influence the training? Peaks were thus annotated out of their context. A trained spectroscopist will refine its peak picking according to signal found also in other spectra. So why not having used corresponding peak-lists (when available at the BMRB) together with spectra data (or possibly reanalyze entire data sets) to obtain ground truth data to train the peak picker ?

Reviewer #3 (Remarks to the Author):

Klukowksi et al describe a platform for fully automated analysis of protein NMR resonance assignments and structures from solution NMR spectra. The platform integrates tools previously developed by the Guentert laboratory for automated resonance assignment, NOESY peak list assignments, and structure generation, together with new algorithms and machine learning modules for automated peak picking, extension/refinement of resonance assignments, and model selection/ refinement. The data analysis process is reduced from weeks to hours. The integrated system has been tested using carefully-organized and curated peak list data for 100 proteins and 1,329 multidimensional NMR spectra, assessing the resulting models against models determined by experts.

This work represents a significant advance in the field of automated analysis of small protein resonance assignments and structures from NMR data. It provides an

automated, reliable, robust, and accurate platform for obtaining assignment and structures from good quality solution NMR data. The platform should lower the barrier to use of solution NMR methods for studies of protein structure, dynamics, and interactions, significantly broadening the impact of these methods. The work is suitable for publication in Nature Communications, subject to adequately addressing the concerns and technical comments outlined below.

1. A key result of this work making it suitable for Nature Communications, are the carefully organized, curated, and referenced peak lists provided for 100 proteins. This is a key result of the work. Although these peak lists are available, along with the resulting assignments and structures, from the author's web site, they should also be made accessible from the public BioMagRes Database. Although not a traditional entry, the zip file of final peak lists, assignments, and structures should be made publicly available through the BioMagRes Database or other appropriate, persistent, publicly-accessible data base.

2. These protein NMR peak list data were provided by a large number of scientists, in part to support exactly the kinds of methods development demonstrated in this work across the community. However, not much credit is given to these scientists. Although these NMR data sets each have their own DOIs (as part of PDB/BMRB depositions), there is no attribution to these scientists except by listing of the corresponding PDB / BMRB ids. The authors should provide a Supplementary Table listing for each of the 100 data sets the PDB/BMRB id's, the complete author lists provided in these entries, the entry titles, and the corresponding DOIs.

3. Related to this point, it appears that 78 / 100 structures were solved by only two labs using similar, standardized data collection protocols. Could this introduce bias into the AI training? The 4D NOESY data sets are largely from one lab. How does this bias the results?

4. Were the data used in this paper processed with the processing scripts deposited in the BMRB along with the data? If so, how was this done; which software was used? Was any additional data processing done, such as phase correction? Can the authors provide recommendations to future efforts to harvest NMR data regarding both the enabling and challenging aspects presented by the data provided for the 100 protein targets that were organized for this paper.

5. Do data sets include both Varian and Bruker, or other types of data? Jeol? What percentage of each? Is the peak picking performance different for Varian, Bruker, Joel data?

6. Although a significant advance outlined in the paper is automated peak picking, in many cases user-curated peak lists were also deposited to BMRB for these proteins. Were the user-curate peak lists used for any of the 100 target proteins? or used for assessment of performance of automated peak picking?

7. The authors mention in the Introduction that dihedral restraints based on chemical shift were generated using Talos_N. Is this analysis automated as part of the Artina process? Or is it done separately and provided as a separate input file? Could the authors discuss how dihedral restraints were curated and included in the calculation. At what stage are they added in the Artina process? Was it repeated as chemical shifts were updated? How were these dihedral restraints assessed in terms of reliability scores, dynamics, and involvement in secondary structures? Or did the authors use the Talos-based restraints deposited with these structures? Please discuss how dihedral restraints were determined and included in the calculation.

8. Many of the reference protein structures were refined with RDC data. How does this impact the accuracy assessments.

- 9. In most cases, the reference structures were energy refined prior to deposition? Would further structural refinement after the Cyana calculation be expected to improve the RMSD to the reference structure? How would restrained energy refinement of the Artina structures affect these results?**
- 10. Did any of the data sets use perdeuterated protein samples? If so, how did this impact the performance?**
- 11. Some of these NMR structures do not have PDB ids. Would it be possible to deposit the two structures to the PDB/BMRB that are not in there yet (this should be required by journal). Is PDB ID 3JZ a typo? It is an X-ray structure.**
- 12. It was not mentioned if there were separated 3D ¹³C-noesy aliphatic and aromatic data and how they were treated? Were they combined? Were any example data sets with simultaneous NC NOESY data used?**
- 13. How was 2D data used? Was it included/used in the training or as a way to filter noise peaks? Were all spectra treated together or just separately for the peak picking/triage process? Further clarification of these details would be useful.**
- 14. Was chemical shift aliasing (folding) considered for the 2D/3D/4D data? If so, how were the folding sweep widths determined? Was it automatic? Was 2D data used? In particular, the 4D CC NOESY is typically collected with small indirect carbon sweep widths of ~21 ppm and indirect ¹H of ~7 ppm, and folding needs to be considered for CA vs CH₃, and for 'unfolding' aromatic peaks. How was this aliasing/folding taken into account for indirect ¹³C and ¹H dimensions?**
- 15. Could there be other reasons that the addition of 4D NOESY data would not lead to improvements in the structural RMSD to the reference? It is surprising that the addition of data with less ambiguity would make structures worse. Doesn't this suggest that treatment of this data should be further investigated/optimized? Perhaps folded peaks were not aliased correctly? Was there a correlation between bb and 'core' sc residue assignment completeness and the RMSD to reference?**
- 16. Several of the target proteins have incomplete chemical shift assignments? When there were poor/missing assignments for aromatic residues in the protein core, did this result in poor RMSD to the reference? Was there a correlation between bb and 'core' sc residue assignment completeness and the RMSD to reference?**
- 17. Since it was the case that improvements were found in chemical shift assignments upon addition of the NOESY data, where were they found? Was it for particular residues types and/or atom types? Was it primarily the aromatic chemical shift assignments that were improved?**
- 18. Did the Artina process make and use stereospecific assignments prochiral methylenes and isopropyl methyl resonances? How did these automatic stereo-specific assignments compare to the stereospecific assignments provided in the BMRB file? Would the accuracy of these assignments be expected to impact structural accuracy? Were stereospecific assignments of isopropyl methyl's considered in the RMSD calculations?**
- 19. How were RMSDs calculated between the two sets of 20 ensembles; the table just reports a single value RMSD for bb or heavy atom between the two sets. Is this a average pairwise RMSD, or the RMSD between "representative" or medoid conformers of each ensemble?**
- 20. The descriptor "end-to-end" automatic structure determination may not be used correctly in this manuscript. It has a specific meaning in AI-land that all automatic steps**

from input to output are controlled and adjustable by the AI, and not just the two modules described in this paper.

Response to Reviewers

In response to the remarks by the reviewers, we have prepared a revised version of the manuscript, as described in the following point-to-point response to the reviewers (remarks by the reviewers in *blue italics*):

Reviewer #1 (Remarks to the Author):

The manuscript describes the AI-based automatic determination of protein structures of proteins with a molecular weight of up to 20kDa in a completely automated manner without human expert interference between recording of NMR spectra to structure.

This is a truly remarkable achievement, unmet by previous extensive efforts to solve this most outstanding NMR protein structure determination problem. As such, the manuscript is of highest interest and quality so that it should be published essentially as is. There are two minor comments, one of which I would really like to see conducted, the second is more stylistic.

We thank the reviewer for this very positive assessment.

Minor comments:

1.) The authors should not only provide the pdb codes for the proteins used but also their name and function and also where structures have been published for these proteins, if the structures have been published and not simply deposited.

We have added the new Supplementary Table 3 that provides this information.

2.) Stylistic comment: On page 4, the authors write: "...to reproduce the ground truth protein structures". With a tinkle in the eyes, I confessed that I do not know what a ground truth structure is. Maybe they want to say: bona fide correct structure?

We agree with the reviewer that the structures determined previously by manual methods are not necessarily perfect and may sometimes differ from the "true native structure" of the protein in solution. To indicate this fact, we now put "ground truth" in quotation marks on p. 3. In machine learning, "ground truth" refers to the reference result that the machine learning model aims to achieve.

Reviewer #2 (Remarks to the Author):

The work presents a machine-learning based approach (named ARTINA) to automatically determine protein structure from processed NMR spectra. The method is built around a new peak-picking procedure and two already existing method for automated chemical shift assignment (FLYA) and structure calculation (CYANA). The

main originalities of the approach are i) the use of deep residual neural networks for NMR peaks identification in multi-dimensional spectra and deconvolution and ii) a fully integrated pipeline, with several cycles of assignments and structure calculation with FLYA and CYANA. For training of the neural networks and validation of the performances of the approach, a benchmark dataset was constructed, that includes 1329 processed NMR spectra from 100 proteins, along with the corresponding chemical shifts (BMRB) and atomic coordinates (PDB) records. The two main criteria chosen by the authors to evaluate the efficacy of ARTINA are i) the percentage of correctly assigned chemical shifts when compared to BMRB records and ii) the closeness of structure solutions from CYANA to the PDB records (measures as a RMSD). Overall, ARTINA correctly assigns more than 90% of the chemical shifts and the median backbone RMSD of the final structures with the reference coordinates is below 1.5 Å, across the whole benchmark set. Additionally, the authors evaluated the impact of the recycling between FLYA and CYANA and also the usefulness of including 4D NOESY spectra.

First, I would like to congratulate the authors on having tackle (and succeeded) in establishing such a benchmark dataset that was definitely missing in the field (although they were some attempts from the CASD initiative and a substantial amount of spectra already available from BMRB as time-domain data, mainly deposited by structural genomics consortia).

The manuscript is sound and well written. The webserver implementation of ARTINA is functional (tested on one dataset provided by the authors on the nmrtist.org website). Knowing the excellent track record of the Riek and Güntert groups in this field, I have no doubt that ARTINA will make the life of many NMR spectroscopists much easier.

As an NMR spectroscopist myself, I'm convinced that this work represents a great advance in the field of protein NMR. But, with the new developments of artificial intelligence algorithms we are currently witnessing in biological science, and especially the recent breakthroughs from Deepmind's AlphaFold or David Baker's RosettaFold in structure prediction, I (provocatively) will question the relevance of an end-to-end NMR analysis pipeline (despite being apparently the best one) ? Indeed, it has recently been shown that AlphaFold can generate models with similar or even better fit to NMR data than for the experimental NMR or X-ray structures (Terejo et al, 2022 <https://doi.org/10.1101/2022.03.09.483701> and Huang et al., 2022 <https://doi.org/10.1002/prot.26246>). Maybe the future of NMR lies more in dynamics analysis, ligand screening, interaction mapping rather than in structure determination ? The authors must discussed more the real advances of their method in this new context of structural biology, a.k.a. the post-AlphaFold era.

We agree that AlphaFold is a game-changing breakthrough in structural biology, and that routine structure determinations of proteins by experimental methods will in many cases no longer be necessary. Nevertheless, there remains space for the experimental methods, e.g., regarding different states of proteins under different conditions or in dynamic exchange or regarding interactions with ligands. Regarding ARTINA, one should keep in mind that our system will be of great use for virtually any biological NMR studies that require the analysis of multidimensional NMR spectra and chemical shift assignments. We have added a new paragraph at the end of the Discussion section on p. 9 to discuss these points.

However, before considering the manuscript for possible publication, I have the following main concerns (that I will detail more below):

- the incomplete methodological details*
- the lack of comparison of the new peak-picker (which lies at the core of the ARTINA workflow) to other existing software with the same purpose*
- the possible biases in training and inference*
- the lack of a "reliability" score for structure proposals*
- and most importantly:*
- the unavailability of source code and trained models from the deep-learning parts*

1) The description of the methods is not complete, i.e. not self-sufficient to reproduce the results (or re-implement the presented algorithm):

Schematics of network architectures for pp-ResNet, deconv-ResNet and the DeepGNN used for expected chemical shifts must be presented since their description is lacking some important details.

pp-ResNet, deconv-ResNet, DeepGNN and GBT are available for download in binary form, together with: architecture schemes, example input data, model input specification, and Python scripts that make predictions with example input data:

- Source code public repository: <https://github.com/PiotrKlukowski/ARTINA>
- pp-ResNet files: https://nmrtist.org/static/public/publications/artina/models/ARTINA_peak_picking.zip
- deconv-ResNet files: https://nmrtist.org/static/public/publications/artina/models/ARTINA_peak_deconvolution.zip
- GNN files: https://nmrtist.org/static/public/publications/artina/models/ARTINA_shift_prediction.zip
- GBT files: https://nmrtist.org/static/public/publications/artina/models/ARTINA_structure_ranking.zip

Python scripts deposited in the git repository automatically download the files required for their execution from the ARTINA server.

The above files provide a full technical specification of the components developed in the ARTINA project, and allow for their independent use in Python. We have updated the Code availability section on p. 16 accordingly.

The main aim of the manuscript is to describe the ARTINA workflow, while the technical details (model binaries, schemes) are available for download. It should be kept in mind that the general ARTINA workflow is likely here to stay, but individual models and their internal parameters are subject to change and its details therefore of a less fundamental nature.

We already see that the ARTINA web server gets substantial attention from the NMR community. At the time of writing this document, over 1100 additional protein spectra have been uploaded to nmrtist.org from laboratories around the world. If user interest continues at this level, we will be able to release new versions of pp-ResNet and deconv-ResNet in the near future. New models will be trained on substantially larger benchmark dataset, validated in practice by users (on nmrtist.org), and described in detail in a future manuscript.

The authors state that a 5-fold cross validation was applied to assess the performance (using only 80% for training and 20 % for testing). What is thus the RMSD values reported in Supp Table 3, Table 1 and Figure 3 and 4 for each entry? Is this the average over the 5tests ?

The idea behind 5-fold cross-validation is to train 5 models, each using about 80% of the data for training and 20% for testing. The split between training and test set is done such that each protein is present exactly once in the test set. There is no need for averaging. Reported RMSDs were obtained directly in the cross-validation experiment. We have updated the text on p. 3 to clarify this point.

In the current implementation available on the server, which training models for pp-ResNet and GBT are used? Was it trained on 100% of the data set?

This question refers to machine learning model finalization. There are several common choices for that step. For example, (a) one can retrain all models using 100% of data, or (b) one can construct a model ensemble by averaging predictions of all 5 models (bagging). In theory, the construction of ensemble improves prediction quality of the ML solution. However, the practical benefits are not always apparent, as bagging increases model prediction time 5-fold. We selected (b) as model finalization approach, and state this now in the Results section on p. 3.

How is calculated the RMSD between structure solutions from ARTINA and reference PDB entries ? Since both reference and query are ensembles of conformation, and a single value is reported for the RMSD to reference, one can question if it is "average vs average of each ensemble" or "average of pairwise RMSD between the target ensemble and the entry ensemble" or else...

RMSDs are between the mean coordinates of the ARTINA structure bundle and the mean coordinates of the corresponding reference PDB structure bundle for the backbone atoms N, C^α, C' (or all heavy atoms). Residue ranges for RMSD calculation (given in Supplementary Table 4) were determined by CYRANGE applied to the region between the first residue of the first secondary structure element and the last residue of the last secondary structure element of the reference PDB structure. We clarified this on p. 3/4 and in a footnote of Supplementary Table 4.

Additionally, in the 2nd round of CASD, a threshold of 1.5 Å was used as criterion to classify a structure solution as accurate when compared to a reference (or ground truth). Here the authors uses 3Å to select entries worth being analyzed more or for the count the overall performance. Can the authors justify this 3 Å criterion ?

We report individual RMSD values for all 100 proteins used in this study, e.g., in Fig. 4, Supplementary Fig. 5, Supplementary Tables 3, 4, 7–10. In addition, we use the (threshold-independent) median RMSD as a measure of the overall performance.

A threshold of 3 Å has been used only for presentation purposes to discuss ARTINA convergence across iterations (Table 1), and to select the most deviating ARTINA structures for detailed analysis in the manuscript.

To avoid possible confusion, we removed the 3 Å threshold from the manuscript, which does not change any of the conclusions, and we extended Table 1 to reporting the number of proteins with RMSD to reference below multiple cutoffs, i.e., 1, 2, 3 Å

The list of BMRB entries used should be given in SI (added to Supp. Table 3 for instance) along with the source of the data and the reference structure (paper citation if there is no PDB and/or BMRB entries)

This information is now provided in the new Supplementary Table 3 (see also Reviewer 1).

TALOS is briefly mentioned as a dependency at the end of the manuscript but its use is not documented. The details on how TALOS was applied must be given in the methods section.

We added the new Methods paragraph 'Torsion angle restraints' on p. 14 to clarify this point.

The GBT (for structure ranking) was trained on the benchmark data set (page 3 “Automated protein structure determination”) using 96 features, including CYANA target function and restraints statistics. This part must be presented in a more detailed fashion because in the current state, it cannot be re-implemented solely from the method description. Especially how the training was done. If the ground truth is PDB record, how the features related to distance restraints was obtained? What are the 96 features ? We need to know that to be sure that there are no circular analysis (GBT can select good structure only from the benchmark ?). This question is also related to my other points about having a possible bias in the benchmark since 80% of the PDB records were originally calculated by CYANA

As pointed out above, details of the models, including GBT and its input features specification are now available for download. There is neither bias nor circular analysis in the GBT model training. We added a paragraph to the Methods section on p. 15 that clarify these issues and explains why details related to PDB deposition are irrelevant for GBT.

*2) I am surprised not to see any comparisons between the new peak-picker (pp-Resnet) and other automated peak-picking approaches. The authors should compare the performance (at least for a subset of the benchmark) with some of the popular algorithms available for automated peak-picking among NMRnet, CCPN, NMR-ViewJ, PICKY, DEEP Picker, CYPICK, AUTOPSY, CV-Peak Picker....
In addition, DEEP Picker, the most recent peak-picker (with deconvolution) also based on deep-learning must be cited since the approach share some similarities with ppResNet and deconv-Resnet (Li et al, 2021 <https://doi.org/10.1038/s41467-021-25496-5>)*

A comprehensive comparison of this kind is, in principle, of interest but beyond the scope of our manuscript and not necessary to support any of our results and conclusions.

It is difficult to compare results of different peak picking models directly, i.e., by comparing and assessing the peak lists produced by different approaches, because, in contrast to BMRB chemical shift assignments and PDB structures, the corresponding ground truth in the form of manually prepared peak lists is publicly available only for a small fraction of the 1329 spectra in this study. In addition, for manual assignment and restraint collection, it is perfectly valid to enter into a peak list only peaks that are relevant for the task at hand (for instance, only one of a pair of transposed cross peaks). Similarly, peaks that are present but could not be interpreted may have been removed from peak lists used in the manual approach. This renders direct quantitative comparison with automatically picked peak lists difficult, since differences do not necessarily indicate shortcomings of either method.

In our manuscript, we do not claim superiority of our peak picking model over other approaches. Our point is to show that the quality of the peak lists produced by our models are sufficient to obtain chemical shift assignments and three-dimensional structures of quality comparable to those from manual approaches. In principle, it is conceivable that the ARTINA workflow could employ other peak picking methods. This would, however, not invalidate any of the results and conclusions of this paper.

We would like to stress that pp-ResNet is a model that (a) handles 2D/3D/4D spectra, (b) has no tuneable parameters, (c) offers full automation (an NMR spectrum can be converted to a peak list by a single command-line application call, without visually looking at the spectrum), (d) is high-throughput – it takes less than 10 minutes to analyse a large NOESY spectrum – and is therefore suitable for large-scale experiments with over 1,300 spectra.

To compare pp-ResNet with any other model, the latter must at least satisfy conditions (a), (b) and (c).

Notable peak picking approaches published in the last 5 years comprise CYPICK (2017), NMRNet (2018), and DEEP Picker (2021).

- DEEP Picker handles only 1D/2D spectra (a). Therefore, it is of limited use for macromolecular structure solving with NMR. Additionally, DEEP Picker has tuneable parameters (b), which have to be calibrated by the user for each spectrum individually. Therefore, it doesn't offer the full automation required in the ARTINA workflow.
- CYPICK handles 2D-4D spectra (same as ARTINA), but requires the user to tune one parameter per spectrum dimension (b). Therefore, it is not suitable as a building block for ARTINA.
- NMRNet fulfils the above conditions and could in principle be compared with pp-ResNet. However, it was developed by one of us and is obviously an inferior predecessor of the peak picking models in ARTINA that features a less developed architecture, lacks deconvolution, and was trained on a comparatively tiny training data set. A detailed study of how much better ARTINA is than its outdated predecessor is of relatively low interest and has no direct relevance to the principal subject matter of this paper, namely the presentation of ARTINA as a fully automated method for NMR spectra analysis. We therefore prefer, in order to not distract readers, to not include such a comparison into the present paper.

We have also added the missing reference to DEEP Picker on p. 2.

3) 80% of protein in the benchmark come from the North East Structural Genomics (NESG) consortium. I suspect this would introduce a bias related to data collection and processing practices? The authors should show their approach perform as good with non-NESG as with NESG entries. Possibly re-train on NESG entries only and predict on the remainder (nonstructural genomics entries and RIKEN and CESG entries)

See our answer to a similar question by Reviewer 3 (Point 3.) below.

I counted 67 PDB entries from the data set (~70%) that employed CYANA for the structure calculation of the deposited coordinates. Would that too introduce a bias, possibly in the GBT ranking? Authors must consider that, evaluate if accuracy is the same in the CYANA and non-CYANA reference structure. Possibly re-train on CYANA and predict on non-CYANA entries?

According to PDB depositions (see Supplementary Table 2), CYANA was used in the manual structure determinations of 73 proteins and programs from the X-PLOR family (X-PLOR, CNS, Xplor-NIH) for 81 proteins. The large majority of restraint files deposited with the PDB are in X-PLOR format (87 out of 94), suggesting that X-PLOR-type software was used for structure refinement.

Software used to prepare PDB depositions plays no role in GBT training, because PDB depositions aren't used to train machine learning models in the ARTINA workflow. As pointed out above in response to another point raised by the reviewer, we have added a paragraph to the Methods section on p. 15 that clarifies these issues and explains why details related to PDB depositions are irrelevant for GBT.

5) The authors emphasizes that ARTINA is usable by non-expert. If the authors want to put this argument forward, the software must then return associated confidence values at each output steps: for peak-picking, it's already the case with Resnet; for assignments, FLYA reports some statistics and also strong assignments, but a confidence estimate is missing for structure proposals. If the whole process is fully automated, with no human intervention, it is necessary to guide for the user to choose between the solutions proposals, especially when they are significantly different. I'm surprised the authors didn't use the predicted resolution estimator that they published some years ago (Bagaria et al. Comput Biol Chem. 2013).

Thank you for this question. It suggests that we should explain more carefully the idea behind ARTINA structure proposals.

ARTINA uses GBT to rank structure proposals by their expected quality. We don't ask the user "to choose between the solution proposals". The model selected as TOP-1 by GBT constitutes the final ARTINA structure. TOP-1 models were used to calculate quality metrics reported in the manuscript.

The <https://nmrtist.org> web server displays a second model (TOP-2) just to present visually possible discrepancies between the top 2 solutions produced by ARTINA. If there are high deviations, the ARTINA output (TOP-1) should be treated with extra caution, especially regarding the protein regions where the discrepancy is present.

Nevertheless, in the response to this comment, we introduced the new pRMSD estimate of structure accuracy (see below; Methods section on p. 15, Fig. 4, Supplementary Table 5), which is computed on the basis of the TOP-1 and the 9 other structure proposals.

The approach of Bagaria et al. (2013) was designed to analyse protein structures irrespectively of the method that was used to solve them. Our GBT model was designed specifically for ARTINA and NMR. The most notable differences are:

- Bagaria et al. use as input only the PDB structure. The GBT model uses also intermediate data produced by ARTINA, such as the list of distance restraints. For example, the number of long-range distance restraints assigned in automated NOE assignment is an important feature that provides information about the quality of the resulting protein structure.
- Bagaria et al. analyse proteins independently from each other, which is more challenging than comparing different structure proposals for the same protein. While working on ARTINA, we found that pairwise comparisons yield more accurate protein rankings. The main reason is that in pairwise comparisons we compare two structure calculations of the same protein. Therefore, quantities, such as the number of restraints or the CYANA target function value, can be compared directly.

I thus strongly encourage the authors to include one of the two following metrics to provide a reliability score associated with the structure proposal.

- ANSURR scores as an independent measure of accuracy (Fowler et al., 2020 <https://doi.org/10.1038/s41467-020-20177-1>). I would also be delighted to see ANSURR score computed for reference and for ARTINA solution in the presented benchmark. (RMSD does not tell if a structure is as good or better or worse, just if it's structurally close or not)

- DP-score to estimate how well the solution structures explain the NOESY data (as main source of restraints) (Huang et al. 2015 doi:10.1007/s10858-015-9955-2)

We have calculated ANSURR and DP (RPF) scores for all 100 proteins and report the results in the new Supplementary Tables 6 and 7. We see no obvious correlation between ANSURR scores and structure accuracy as measured by the RMSD to PDB reference. Also, the DP scores of the 100 proteins do not correlate strongly with the RMSD to reference.

The example of the 2M47, for which the best ranked solution is 4.7 Å from the reference but the 2nd one is much closer, is very interesting. On that point, the authors concluded that automated structure determination is unstable for this protein. But a few years ago, the group of P. Guntert published a new approach name consensus calculation (Buchner L et al. 2015 doi: 10.1016/j.str.2014.11.014.) that was designed exactly to cope with such situations. If the authors decided not to implement the consensus approach as a final step of the pipeline, they should discuss why.

We have calculated consensus structure bundles for all 100 proteins by the method of Buchner et al. and report the results in the new Supplementary Table 8.

In the case of 2M47, the ARTINA structure bundle has an RMSD to mean of 1.59 Å and an RMSD to PDB reference of 4.72 Å, whereas the consensus structure bundle has an RMSD to mean of 2.39 Å and an RMSD to PDB reference of 3.63 Å, i.e., the consensus structure bundle is somewhat closer to reference and its RMSD to mean is a better indicator of structure accuracy than the ARTINA structure. Obviously, the GBT model missed to select the best out of 10 structure proposals, since the second structure proposed by GBT has a much lower RMSD to reference of 1.66 Å.

Nevertheless, we prefer to not include the consensus structure calculation approach into the standard ARTINA workflow because (a) it is very computation-intensive (~20 times more computation time for the structure calculation), and (b) we have devised an alternative predicted RMSD to reference value that correlates well with the actual RMSD to PDB reference and can be obtained without additional structure calculation from the 10 structure proposals that are anyway calculated in ARTINA.

The new predicted RMSD (pRMSD) is introduced on p. 4 and defined in the Methods section on p. 15. Results are given in Fig. 6 and Supplementary Table 4. The average deviation between actual and predicted RMSDs for the 100 proteins in this study is 0.35 Å. In no case, the true RMSD exceeds the predicted one by more than 1 Å.

4) ARTINA can be used on a webserver. However, it is not possible to download the software. The code for inference of structure from spectra (end-to-end) must be made available in a repository. If some pieces of software have closed licensing (I think about CYANA for instance), then it would simply become a requirement and it would not be mandatory to provide it (but rather let to the user to make sure it is installed properly, following the indications of the code repository). We are now in the era of open science and even Google (a definitely "for profit" company!) made the code and models of its structure prediction AlphaFold available for free with a creative common license. I acknowledge the effort of the authors for making the results from their benchmark available but if other groups wants to build on the advances from ResNet or the GBT for instance, those must be available (at least in binary code, the best would be open source).

We provide serialized pp-ResNet, deconv-ResNet, DeepGNN and GBT models, together with prediction scripts (Python), exemplary input data, and the technical description of the models. These files allow other groups to build on the advances of this work (see response to question 1 above).

Other remarks and/or questions:

- It is said that ARTINA correctly assigned 90.39 % of chemical shifts. Are all assignment or only the ones classified as strong used to assess the accuracy ?

All (strong and weak) assignments made by FLYA are compared with chemical shifts deposited in BMRB. We have clarified this in the text on p. 4.

- How would this work on dimeric protein ? Also, I noticed that two entries in the benchmark are in fact protein/RNA complexes. It would be interesting to see the amount of RNA peaks that could not possibly be assigned and how ARTINA did cope with that ? It would also an infesting point to add to the discussion.

These are interesting questions that we plan to address in a future release of ARTINA. The present, first release of ARTINA can handle only monomeric proteins. Therefore, we don't refer to RNA spectra analysis/assignment in the manuscript.

- Page 5, 1st paragraph, line 6: "Nevertheless, the residues contributing to the high RMSD value are distributed more extensively than in 2L82 and 2M47 just discussed." 2M47 should be 2KCD instead.

Thank you for this comment. We have corrected this mistake in the manuscript on p. 5.

- ARTINA has no free parameters and human intervention is not required. But what if some additional NMR data are available (RDC, J-couplings, H-bond from HDX) or possibly an initial model (possibly from AlphaFold)? How could it integrated in the ARTINA pipeline ?

These are obvious extensions of the basic ARTINA workflow that we plan to address in a future release. We have mentioned these points in the Discussion on p. 8/9.

- Other webservers exist for chemical shift assignment and/or NMR structure calculation . Some of them must be cited (on the top of my head, the ARIAweb and I-Pine webservers).

We included references to these methods on p. 2.

- For the training of pp-ResNet, manual annotation of 670 000+ 2D spectrum fragments was necessary. To me, this step include some subjectivity (conservative or aggressive annotation?). How does it influence the training? Peaks were thus annotated out of their context. A trained spectroscopist will refine its peak picking according to signal found also in other spectra. So why not having used corresponding peak-lists (when available at the BMRB) together with spectra data (or possibly reanalyze entire data sets) to obtain ground truth data to train the peak picker ?

The role of pp-ResNet is to extract visual information from the spectrum. The model constructs lists of cross-peaks that visually resemble true peaks rather than artefacts. This assessment is made for each cross-peak independently. A certain fraction of peaks is ambiguous, which is an integral aspect of any peak picking (automated or manual). At this stage, ARTINA tends to retain ambiguous peaks (usually with lower classifier response value) for further analysis. This is desired, and the visual layer of ARTINA was specifically designed to follow these principles.

In the later stages of the ARTINA workflow, all peak lists are analysed by FLYA. Here, peak lists are refined according to signals found also in the other spectra. Finally, ARTINA has 3 iterations, where peak positions are

reassessed given preliminary protein structures. Therefore, ARTINA goes 3 times from cross-peak positions to chemical shifts and the structure, refining and reinterpreting cross-peaks each time. Thus, we deal with peak ambiguity at later stages of data analysis, not during predictions with pp-ResNet.

Following reviewer 2's analogy to a trained spectroscopist, pp-ResNet is used to "look at" a specific point in the spectrum. Cross-checking spectra (signals observed in other spectra, signals aligned vertically/horizontally) corresponds to higher-level analysis, because it involves logic and knowledge about expected peaks. This logic is implemented in FLYA chemical shift assignment, and NOE assignment with structure calculation.

There are several problems with the use of deposited manual prepared peak lists for the peak picking model training and evaluation. Perhaps, they are the reason why no machine learning peak picker was published that uses them in practice. Even the recent DEEP Picker by Li et al. (2021) used generated data rather than deposited peak lists.

The reasons, why we didn't use deposited peak lists in ARTINA are:

- Predominantly NOESY peak lists have been deposited with the BMRB, other spectrum types being largely underrepresented.
- Refined peak lists are something that one expects to get in the end of the process, after cross-checking different NMR spectra and making assignments. If a human cannot classify a signal purely by looking at the spectrum, we cannot expect the pp-ResNet model to do it, as it receives only visual information as input.

One should also keep in mind that both positive and negative examples are required to train the model. Even if the above problems were overcome, it would help only with the annotation of positive examples. Negative examples would still need to be selected manually or semi-manually.

Peak picking is an intrinsically ambiguous process. Since the goal of ARTINA is to deliver assignments and structures, we focused primarily on making unambiguous comparisons of chemical shift assignments (BMRB depositions) and structures (PDB).

Reviewer #3 (Remarks to the Author):

Klukowski et al describe a platform for fully automated analysis of protein NMR resonance assignments and structures from solution NMR spectra. The platform integrates tools previously developed by the Guentert laboratory for automated resonance assignment, NOESY peak list assignments, and structure generation, together with new algorithms and machine learning modules for automated peak picking, extension/refinement of resonance assignments, and model selection/ refinement. The data analysis process is reduced from weeks to hours. The integrated system has been tested using carefully-organized and curated peak list data for 100 proteins and 1,329 multidimensional NMR spectra, assessing the resulting models against models determined by experts.

This work represents a significant advance in the field of automated analysis of small protein resonance assignments and structures from NMR data. It provides an automated, reliable, robust, and accurate platform for obtaining assignment and structures from good quality solution NMR data. The platform should lower the barrier to use of solution NMR methods for studies of protein structure, dynamics, and interactions, significantly broadening the impact of these methods. The work is suitable for publication in Nature Communications, subject to adequately addressing the concerns and technical comments outlined below.

1. A key result of this work making it suitable for Nature Communications, are the carefully organized, curated, and referenced peak lists provided for 100 proteins. This is a key result of the work. Although these peak lists are available, along with the resulting assignments and structures, from the author's web site, they should also be made accessible from the public BioMagRes Database. Although not a traditional entry, the zip file of final peak lists, assignments, and structures should be made publicly available through the BioMagRes Database or other appropriate, persistent, publicly-accessible data base.

In addition to publication of this data on <https://nmrtist.org/>, we will make the same data available at the permanent ETH Research Collection (<https://www.research-collection.ethz.ch/>).

2. These protein NMR peak list data were provided by a large number of scientists, in part to support exactly the kinds of methods development demonstrated in this work across the community. However, not much credit is given to these scientists. Although these NMR data sets each have their own DOIs (as part of PDB/BMRB depositions), there is no attribution to these scientists except by listing of the corresponding PDB / BMRB ids. The authors should provide a Supplementary Table listing for each of the 100 data sets the PDB/BMRB id's, the complete author lists provided in these entries, the entry titles, and the corresponding DOIs.

We have added the new Supplementary Table 3 that provides this information (see answer to Reviewer 1).

3. Related to this point, it appears that 78 / 100 structures were solved by only two labs using similar, standardized data collection protocols. Could this introduce bias into the AI training? The 4D NOESY data sets are largely from one lab. How does this bias the results?

A similar question was asked by Reviewer 2.

This is a good point. We are aware that a large fraction of our dataset comes from NESG. Although we took a substantial effort to collect the benchmark datasets from several acquisition channels, the highest number of spectra were obtained from NESG via the BMRB FTP server.

It can indeed not be ruled out that the large fraction of proteins from the NESG structural genomics program could have introduced a bias that may slightly affect the performance of ARTINA on proteins from other laboratories. However, such bias will be temporary because, from now on, continuous data collection by the nmrt-ist.org webserver will soon supersede the present concentration on NESG structures and allow for retraining the models with more diversity in the training data set. At the time of writing this document, we collected over 1100 spectra in addition to the 1329 stored in the benchmark for this paper. We have added this point to the Discussion on p. 8.

An assessment of possible bias can be obtained by aggregating proteins by the origin of spectra (NESG vs. others) and then comparing quality metrics of automated protein structure determination (e.g., the RMSD to reference). We did such an analysis, demonstrating that on average ARTINA performs equally well on NESG proteins (median RMSD to reference 1.44 Å) and non-NESG proteins (1.42 Å). We added this point on p. 4.

4. Were the data used in this paper processed with the processing scripts deposited in the BMRB along with the data? If so, how was this done; which software was used? Was any additional data processing done, such as phase correction? Can the authors provide recommendations to future efforts to harvest NMR data regarding both the enabling and challenging aspects presented by the data provided for the 100 protein targets that were organized for this paper.

We added the new Methods paragraph 'Spectrum benchmark collection' on p. 9/10 to clarify this point.

5. Do data sets include both Varian and Bruker, or other types of data? Jeol? What percentage of each? Is the peak picking performance different for Varian, Bruker, Jeol data?

This information is included in the new Supplementary Table 3, which was prepared on the basis of the PDB files. Overall, the PDB files state that both Bruker and Varian spectrometers were used for 58 (mostly NESG) proteins, exclusively Bruker for 31, and exclusively Varian for 7 proteins. Use of JEOL spectrometers is not reported in the PDB files. We did not see remarkable differences for different spectrometer types. We have stated this on p. 4.

6. Although a significant advance outlined in the paper is automated peak picking, in many cases user-curated peak lists were also deposited to BMRB for these proteins. Were the user-curated peak lists used for any of the 100 target proteins? or used for assessment of performance of automated peak picking?

See our answer to a similar question by Reviewer 2.

7. The authors mention in the Introduction that dihedral restraints based on chemical shift were generated using Talos_N. Is this analysis automated as part of the Artina process? Or is it done separately and provided as a separate input file? Could the authors discuss how dihedral restraints were curated and included in the calculation. At what stage are they added in the Artina process? Was it repeated as chemical shifts were updated? How were these dihedral restraints assessed in terms of reliability scores, dynamics, and involvement in secondary structures? Or did the authors use the Talos-based restraints deposited with these structures? Please discuss how dihedral restraints were determined and included in the calculation.

We added the new Methods paragraph 'Torsion angle restraints' on p. 14 to clarify this point.

8. Many of the reference protein structures were refined with RDC data. How does this impact the accuracy assessments.

According to the restraint files deposited with the PDB, RDCs were used for the manual structure determinations of 9% of the proteins (2KCT, 2KD1, 2KKZ, 2KZV, 2LFI, 2LGH, 2LL8, 2LML, 2L9R; in all cases only N-HN RDCs). The median RMSD to reference for these structures (1.53 Å) is similar to the that for all 100 proteins

(1.44 Å). The present version of ARTINA does not use RDC data. It is conceivable that a future implementation of automated RDC analysis could improve the accuracy of the structures.

9. In most cases, the reference structures were energy refined prior to deposition? Would further structural refinement after the Cyana calculation be expected to improve the RMSD to the reference structure? How would restrained energy refinement of the Artina structures affect these results?

To address this question, we performed energy refinement in explicit water for all 100 proteins. The results, given in the new Supplementary Table 9, show that energy refinement does not affect (positively or negatively) the accuracy of the structures. Backbone RMSDs to the reference PDB structure before and after energy refinement exhibit an average difference of 0.05 Å, a maximal difference of 0.16 Å, and a linear correlation coefficient of 0.998.

10. Did any of the data sets use perdeuterated protein samples? If so, how did this impact the performance?

Perdeuteration was not used for the 100 proteins in the benchmark. Almost all protein samples used uniform ^1H , ^{13}C , ^{15}N labelling. Protein 6SOW was only 20% ^{13}C -labeled (see also Supplementary Fig. 4).

11. Some of these NMR structures do not have PDB ids. Would it be possible to deposit the two structures to the PDB/BMRB that are not in there yet (this should be required by journal). Is PDB ID 3JZ a typo? It is an X-ray structure.

PDB/BMRB depositions should be made by the spectroscopist who measured the spectra and studied the protein. We agree with the reviewer's opinion on PDB deposition, but don't see us in a position to make these depositions in place of the original authors.

We thank the reviewer for pointing out the misleading issue with 3JZK. This accession code points to the X-ray structure of the protein MDM2. We used NMR spectra of this protein for ARTINA. The corresponding conventional NMR structure has, however, not been deposited with the PDB. To avoid confusion, we have renamed this protein in the revised manuscript to the non-PDB name MDM2.

12. It was not mentioned if there were separated 3D ^{13}C -noesy aliphatic and aromatic data and how they were treated? Were they combined? Were any example data sets with simultaneous NC NOESY data used?

Three approaches to 3D ^{13}C -NOESY spectra measurement are represented in the benchmark dataset, as explained in a new Methods paragraph on p. 9/10.

13. How was 2D data used? Was it included/used in the training or as a way to filter noise peaks? Were all spectra treated together or just separately for the peak picking/triage process? Further clarification of these details would be useful.

All spectra types (2D, 3D, 4D) were used to train pp-ResNet. We do not differentiate between 2D and 3D/4D spectra in the pp-ResNet prediction. The role of the model is to inspect visually a fragment of an NMR spectrum, without using any external information (such as peak positions in other spectra). The analysis that involves non-local contextual information is performed by FLYA (chemical shift assignment) and CYANA (NOE assignment).

14. Was chemical shift aliasing (folding) considered for the 2D/3D/4D data? If so, how were the folding sweep widths determined? Was it automatically? Was 2D data used? In particular, the 4D CC NOESY is typically collected with small indirect carbon sweep widths of ~21 ppm and indirect ^1H of ~7 ppm, and folding needs to be considered for CA vs CH₃, and for 'unfolding' aromatic peaks. How was this aliasing/folding taken into account for indirect ^{13}C and ^1H dimensions?

ARTINA can automatically unalias (unfold) signals in 2D/3D/4D spectra. We have added a new subsection to the Methods section on p. 12/13, which explains also the automatic unfolding of heavily overlapped 4D CC-NOESY spectra by reference to corresponding unfolded 3D ^{13}C -NOESY peak lists.

15. Could there be other reasons that the addition of 4D NOESY data would not lead to improvements in the structural RMSD to the reference? It is surprising that the addition of data with less ambiguity would make structures worse. Doesn't this suggest that treatment of this data should be further investigated/optimized? Perhaps folded peaks were not aliased correctly? Was there a correlation between bb and 'core' sc residue assignment completeness and the RMSD to reference?

We believe that there are several reasons, why the addition of 4D NOESY (which, in our benchmark, was always supplementary to 3D NOESYs) did, on average, not lead to improvements of the RMSD to the reference structure:

- Signals recorded in 4D NOESYs are strongly folded, such that regular and folded signals both overlap and have the same sign of the amplitude. Although 4D NOESY in principle reduces the ambiguity of NOE assignment, strong folding has a counter effect.
- Several 4D NOESY were measured on small systems, e.g., 2HEQ (84 residues), 2K3D (87), 2K75 (106), 2KBN (109), 2KCD (120). Usually, 3D NOESY provide enough information to reliably determine protein structures in this size range. The information from 4D NOESY is then largely redundant.
- Proteins in the benchmark dataset typically have more than 10 spectra for the sequence-specific chemical shift assignment, which reduces ambiguity of chemical shift assignment by data redundancy.

We don't think that unfolding ambiguity was a major problem. The large majority of 4D NOESY peaks could be unfolded by reference to corresponding peaks in 3D ^{13}C -NOESY. See also our answer to a previous question.

We included an analysis of the RMSD in context of chemical shift assignment accuracy of core residues (Supplementary Table 10). Although proteins with accurately assigned side-chains of core residues tend to have lower RMSD (Pearson's correlation coefficient -0.58), this factor alone is insufficient to predict the quality of the protein fold. The correlation between accuracy of backbone and side-chain shift assignment for core residues is 0.62. The correlation between the accuracy of the backbone core assignment and the RMSD is -0.43 .

16. Several of the target proteins have incomplete chemical shift assignments? When there were poor/missing assignments for aromatic residues in the protein core, did this result in poor RMSD to the reference? Was there a correlation between bb and 'core' sc residue assignment completeness and the RMSD to reference?

To answer this question, we selected residues with solvent-accessible surface area (SASA) in the protein of less or equal than 20% of the corresponding SASA of the isolated residue. Subsequently, we calculated the chemical shift assignment accuracy respectively for each core residue type (Supplementary Table 10), as well as aggregated accuracy scores for: core residue backbone chemical shifts (Pearson's correlation coefficient -0.43 with protein RMSD), core residue side-chain chemical shifts (-0.58), aromatic core residues all shifts (-0.37), and core residues all chemical shifts (-0.57).

17. Since it was the case that improvements were found in chemical shift assignments upon addition of the NOESY data, where were they found? Was it for particular residues types and/or atom types? Was it primarily the aromatic chemical shift assignments that were improved?

To answer this question, we calculated the chemical shift assignment accuracy with respect to each amino acid type (Supplementary Table 13 for all input spectra, Table 14 for all spectra except NOESY). The results suggest that the availability of NOESY experiments consistently improves chemical shift assignments of all residue types. The improvement is particularly strong for aromatic residues (Phe 61.6% vs. 76.5%, Trp 52.5% vs. 80%, and Tyr 71.4% vs. 89.7%), but not limited to this group. We believe that NOESY experiments have rich information content, which reduces the ambiguity of resonance assignments, yielding the overall positive impact on the accuracy. We have extended the text at the end of the Results section on p. 7.

18. Did the Artina process make and use stereospecific assignments prochiral methylenes and isopropyl methyl resonances? How did these automatic stereo-specific assignments compare to the stereospecific assignments provided in the BMRB file? Would the accuracy of these assignments be expected to impact structural accuracy? Were stereospecific assignments of isopropyl methyl's considered in the RMSD calculations?

The present version of ARTINA does not make stereospecific assignments. The method may be extended for this in the future. In particular, it is conceivable to include stereospecific assignments for isopropyl methyl groups by automated analysis of ^{13}C -HSQC spectra from 10% ^{13}C -labeled protein samples.

19. How were RMSDs calculated between the two sets of 20 ensembles; the table just reports a single value RMSD for bb or heavy atom between the two sets. Is this a average pairwise RMSD, or the RMSD between "representative" or medoid conformers of each ensemble?

See answer to the same question by Reviewer 2. We clarified this on p. 3/4 and in a footnote of Supplementary Table 4.

20. The descriptor “end-to-end” automatic structure determination may not be used correctly in this manuscript. It has a specific meaning in AI-land that all automatic steps from input to output are controlled and adjustable by the AI, and not just the two modules described in this paper.

We agree that the term “end-to-end” may confuse readers and have removed it from the manuscript. It wasn't used in the context of end-to-end training, which would imply that all machine learning models are trained together with the possibility to propagate gradients through all steps of data analysis.

Reviewer #1 (Remarks to the Author):

The authors have adequately addressed my previous minor concerns. The manuscript should now be accepted.

Reviewer #2 (Remarks to the Author):

The authors did an excellent job in addressing the reviewer concerns. This is definitely an important and novel work. I wish to thank the authors for their efforts to make the ML models and data available. I recommend publication of the revised manuscript as is.

Reviewer #3 (Remarks to the Author):

We thank the authors for making extensive and careful revisions comprehensively and effectively address most of the concerns raised by the 3 reviews.

Again - we feel this work is a major advance and should be published in Nature Communications as soon as possible. We congratulate the authors on this breakthrough.

However - there are a few minor points in our Original Review which should be addressed before final acceptance.

General Comment: A key result of this work making it suitable for Nature Communications, are the carefully organized, curated, and referenced peak lists provided for 100 proteins. This is a key result of the work. ... should be made publicly available through the BioMagRes Database or other appropriate, persistent, publicly-accessible data base.

The authors responded: In addition to publication of this data on <https://nmrtist.org/>, we will make the same data available at the permanent ETH Research Collection (<https://www.research-collection.ethz.ch/>).

However – we cannot locate these NOESY peak list and other data. This public, permanent accessibility should be verified by Nature Communication editors before the paper is accepted for publication

Comment 6 regarding pp-Resnet. Were the user-curate peak lists ... used for assessment of performance of automated peak picking?

The authors refer to a related question from Reviewer 2. However, reviewer 2 asked for a comparison of the automated peak picking results with other automated peak pickers; our question was to compare with the manually-refined NOESY peak lists that are available in many of the BMRB depositions.

The authors feel this comparison is outside the scope of this work. We do not agree – since automated peak picking is an essential component of the Artina pipeline, the quality of peak picking should be assessed by some "ground truth" data. However – we do agree that the high quality of the resulting structure models speak for themselves, and if Reviewer 2 also agrees, the paper can be published without this analysis. Hopefully it will be provided in a follow up study.

Table 7 RPF Scores. The lack of correlation between RPF scores and structure accuracy noted in the response to Reviewer 2 contradicts several published studies showing a good correlation between RPF-DP and rmsd or GDT scores. There are a few reasons for this. The range of model accuracies across the final models is relatively small – a stronger correlation is observed when there are significantly inaccurate decoys across the assessment set. Second, the RPF scores are

also impacted by resonance assignment inaccuracies. As we understand, ARTINA chemical shift assignments are generally ~90% accurate, which can affect the DP score. Interestingly, about half of the DP-scores were less than 0.7, while the deposited structures generally have DP > 0.7 (some as high as 0.9), suggesting that the match between the model structure / chemical shifts and the data isn't that good, and could be improved. This point need not be addressed in the text, but could be mentioned in a footnote to Suppl Table 7.

Gaetano Montelione
Theresa Ramelot

Response to Reviewers

In response to the remarks by the reviewers, we have prepared a revised version of the manuscript, as described in the following point-to-point response to the reviewers (remarks by the reviewers in *blue italics*):

Reviewer #1 (Remarks to the Author):

The authors have adequately addressed my previous minor concerns. The manuscript should now be accepted.

We thank the reviewer for this positive assessment.

Reviewer #2 (Remarks to the Author):

The authors did an excellent job in addressing the reviewer concerns. This is definitely an important and novel work. I wish to thank the authors for their efforts to make the ML models and data available. I recommend publication of the revised manuscript as is.

We thank the reviewer for this positive assessment.

Reviewer #3 (Remarks to the Author):

We thank the authors for making extensive and careful revisions comprehensively and effectively address most of the concerns raised by the 3 reviews.

Again - we feel this work is a major advance and should be published in Nature Communications as soon as possible. We congratulate the authors on this breakthrough.

However - there are a few minor points in our Original Review which should be addressed before final acceptance.

General Comment: A key result of this work making it suitable for Nature Communications, are the carefully organized, curated, and referenced peak lists provided for 100 proteins. This is a key result of the work. ... should be made publicly available through the BioMagRes Database or other appropriate, persistent, publicly-accessible data base.

The authors responded: In addition to publication of this data on <https://nmrtist.org/>, we will make the same data available at the permanent ETH Research Collection (<https://www.research-collection.ethz.ch/>).

However – we cannot locate these NOESY peak list and other data. This public, permanent accessibility should be verified by Nature Communication editors before the paper is accepted for publication

As specified in the Data Availability section of the manuscript, this data is available for download from https://nmrtist.org/static/public/publications/artina/ARTINA_results.zip. The same data will be available shortly from the permanent ETH Research Collection at <https://www.research-collection.ethz.ch/handle/20.500.11850/568621>. (DOI 10.3929/ethz-b-000568621). For each of the 100 proteins, this data set contains the peak lists for all spectra (including NOESY), chemical shift assignments, and structures obtained with ARTINA.

Comment 6 regarding pp-Resnet. Were the user-curate peak lists ... used for assessment of performance of automated peak picking?

The authors refer to a related question from Reviewer 2. However, reviewer 2 asked for a comparison of the automated peak picking results with other automated peak pickers; our question was to compare with the manually-refined NOESY peak lists that are available in many of the BMRB depositions.

The authors feel this comparison is outside the scope of this work. We do not agree – since automated peak picking is an essential component of the Artina pipeline, the quality of peak picking should be assessed by some "ground truth" data. However – we do agree that the high quality of the resulting structure models speak for themselves, and if Reviewer 2 also agrees, the paper can be published without this analysis. Hopefully it will be provided in a follow up study.

We agree that a comparison of the user-curated peak lists, as far as they are available (i.e., for a small fraction of the 1329 spectra in the benchmark set), with the corresponding ARTINA peak lists could be an interesting topic for a follow-up study. That said, we feel that it would overload the present, already voluminous manuscript that has the different focus of demonstrating that our fully automated method delivers correct assignments and structures for a large number of proteins. Since assignments and structures depend crucially on the peak lists, this would not be possible unless the upstream peak picking had worked well, too.

In our whole study, we have not used manually prepared peak anywhere, neither for the preparation of training data nor for the assessment of results. We have added a sentence to the manuscript on top of p. 10 to clarify this.

Table 7 RPF Scores. The lack of correlation between RPF scores and structure accuracy noted in the response to Reviewer 2 contradicts several published studies showing a good correlation between RPF-DP and rmsd or GDT scores. There are a few reasons for this. The range of model accuracies across the final models is relatively small – a stronger correlation is observed when there are significantly inaccurate decoys across the assessment set. Second, the RPF scores are also impacted by resonance assignment inaccuracies. As we understand, ARTINA chemical shift assignments are generally ~90% accurate, which can affect the DP score. Interestingly, about half of the DP-scores were less than 0.7, while the deposited structures generally have DP > 0.7 (some as high at 0.9), suggesting that the match between the model structure / chemical shifts and the data isn't that good, and could be improved. This point need not be addressed in the text, but could be mentioned in a footnote to Suppl Table 7.

Gaetano Montelione Theresa Ramelot

We agree with the reviewer that the relatively weak (but not completely lacking) correlation between structure accuracy (as quantified by the RMSD to the reference structure) and DP-scores can be due to the fact that, for the almost all of the 100 proteins we are looking at slight differences in structure accuracy (RMSDs in the range 1–3 Å) rather than at detecting incorrectly folded structures, and that differences in chemical shift assignments could play a role. I would expect that user-curated, refined NOESY peak lists are “cleaner” than those used as input for the ARTINA structure calculation. This should increase DP-scores for the manual lists but would not affect the quality of the ARTINA structure, since the algorithm has the possibility to discard artifact NOESY peaks. This could be investigated in the above-mentioned possible follow-up study using manually prepared NOESY peak lists for comparison.

For the present manuscript, we have extended the footnote of Supplementary Table 7 to provide more information on the definition of the different RPF scores and to point out that they depend on the NOESY peak lists, chemical shift assignments, and structures.